# Structure-Aware Random Fourier Kernel for Graphs

**Jinyuan Fang**[1,2]**, Qiang Zhang**[3,4,5]**, Zaiqiao Meng**[6,7]**, Shangsong Liang**[1,2,7*]
[1] School of Computer Science and Engineering, Sun Yat-sen University, China
[2] Guangdong Key Laboratory of Big Data Analysis and Processing, Guangzhou, China
[3] Hangzhou Innovation Center, Zhejiang University, China
[4] College of Computer Science and Technology, Zhejiang University, China
[5] AZFT Knowledge Engine Lab, China
[6] School of Computing Science, University of Glasgow, United Kingdom
[7] Mohamed bin Zayed University of Artificial Intelligence, United Arab Emirates
{fangjy6@gmail.com; qiang.zhang.cs@zju.edu.cn; zaiqiao.meng@gmail.com}
{liangshangsong@gmail.com}

## Abstract

Gaussian Processes (GPs) define distributions over functions and their generalization capabilities depend heavily on the choice of kernels. In this paper, we propose a novel structure-aware random Fourier (SRF) kernel for GPs that brings several benefits when modeling graph-structured data. First, SRF kernel is defined with a spectral distribution based on the Fourier duality given by the Bochner's theorem, transforming the kernel learning problem to a distribution inference problem. Second, SRF kernel admits a random Fourier feature formulation that makes the kernel scalable for optimization. Third, SRF kernel enables to leverage geometric structures by taking subgraphs as inputs. To effectively optimize GPs with SRF kernel, we develop a variational EM algorithm, which alternates between an inference procedure (E-step) and a learning procedure (M-step). Experimental results on five real-world datasets show that our model can achieve state-of-the-art performance in two typical graph learning tasks, i.e., object classification and link prediction.

## 1 Introduction

Gaussian Process (GP) is a typical Bayesian non-parametric method for function learning. Choosing appropriate kernels profoundly improves the generalization performance of GPs on various learning tasks [1, 2]. Stationary kernels, such as Radial Basis Function (RBF) kernel, are common choices of GP kernels and have been successfully applied in many regression and classification problems [3, 4, 5]. However, due to their restricted smoothness assumption and the limited expressiveness, stationary kernels may not be capable for complex data structures, such as graph-structured data [6], where both object features and geometric structures are essential.

To improve GP's performance on data with complex structures, some expressive kernels have been proposed. Two typical expressive kernels are: compositional kernels [7, 8, 9], which are obtained based on the kernel composition rules [10], and deep kernels [11, 12], which are defined as the stack of a deep neural network and a base kernel. Alternatively, the spectral kernels are defined in the spectral domain [2, 13, 14]. Nonetheless, when modeling graph-structured data, prior kernels face several challenges. First of all, the flexibility and expressiveness of these kernels are limited by the simple base kernels upon which they are built, and the optimization of base kernels is computationally expensive [15]. Second, these kernels are usually feature-based, which indicates that they capture the similarities between graph objects based solely on object features. It may be inefficient for them to

---

[*]Corresponding Author.

35th Conference on Neural Information Processing Systems (NeurIPS 2021).

capture the structural smoothness in graph data, i.e., neighboring objects tend to share similar function values [6]. However, capturing such structural smoothness is critical to the success of many graph learning methods [16, 17, 18]. In another line of work, structure-based Laplacian kernels [19, 20, 21] have been proposed to model functions over graphs. They are obtained by applying transformation functions to the graph Laplacian matrices. However, they rarely utilize the object features and may be suboptimal to model complicated functions over large-scale graphs.

To tackle the aforementioned challenges, we propose a **S**tructure-aware **R**andom **F**ourier (SRF) kernel, which is both feature-based and structure-aware. To improve the kernel expressiveness, inspired by the Bochner's theorem [22], we define SRF kernel with a spectral distribution based on the Fourier duality between the kernel and the spectral distribution. This definition brings some benefits: (1) the SRF kernel converts the kernel learning problem into a distribution inference problem, and the resulting SRF kernel will be fairly expressive if the dual spectral distribution is adequately complex; (2) the SRF kernel admits a random Fourier feature formulation for scalable learning, which leads to a linear computational complexity with respect to the number of data points. Compared with feature-based kernels and Laplacian kernels, SRF kernel enables to leverage both object features and local geometric structures to measure similarities by taking subgraphs as inputs. Moreover, to effectively optimize the GPs with proposed SRF, abbreviated as GPSRF, we develop a variational Expectation-Maximization (EM) method [23], which alternates between an inference procedure (E-step) for posterior inference and a learning procedure (M-step) for parameter learning. It's empirically verified that GP with SRF kernel can achieve superior performance than GPs with feature-based kernels or Laplacian kernels when modeling graph-structured data.

Our contributions can be summarized as follows: (1) We propose a novel SRF kernel for modeling functions over graphs, which enables to leverage both geometric structures and object features. (2) Based on the SRF kernel, we propose a GP model, GPSRF, for graph learning tasks. (3) To effectively optimize GPSRF, we develop a variational Expectation-Maximization algorithm. (4) Experimental results on five real-world graph datasets show that GPSRF can outperform strong baselines in two typical graph learning tasks, i.e., object classification and link prediction.

## 2 Related Work

**Gaussian Processes and Kernel Learning for Graphs.** GP models for graph-structured data have been widely studied. They have been proposed for different learning tasks, such as object classification [24, 25, 18], link prediction [26, 27], user profiling [28, 29, 29, 30] and graph classification [5, 31]. Nonetheless, the kernels applied in these methods are feature-based, which may be suboptimal in capturing the local structural smoothness in graph data. In contrast, the structure-based Laplacian kernels [19, 20, 21] capture object similarities based on the geometric structures. However, they rarely utilize the rich object features and may not be capable enough to model the complicated functions over large-scale graphs. Our GPSRF differs from these works in that: (1) the SRF kernel in GPSRF is both feature-based and structure-aware; (2) the random Fourier feature formulation of our GPSRF enables efficient optimization with reduced computational complexity.

Similar to this work, the graph convolutional kernel [32] proposes to incorporate graph convolution network in kernels to leverage local structure information. However, the variational inducing point method [33] used in [32] for posterior inference has a higher computational cost than our GPSRF (see Appendix E.6 for empirical results). In a different manner, the deep spectral kernel network (DSKN) [34] proposes to form an expressive kernel by staking multiple layers of base kernels, while we propose to define an expressive SRF kernel with a complex spectral distribution. The kernel graph convolutional network (KGCN) [35] proposes to learn graph representations with graph kernels and then uses the convolution network to deal with the graph classification task. By contrast, our GPSRF uses graph neural network to obtain SRF kernel and then leverages the Gaussian Process to handle the graph learning tasks.

Another line of research that is closely related to ours is the graph kernels [36], which are kernel functions between pairs of graphs. The graph kernels measure similarities between two graphs for graph classification tasks, whereas our SRF kernel measures similarity between two objects in a single graph for the node-oriented tasks such as object classification and link prediction. Of special interest to us is the neighborhood aggregation based graph kernels such as Weisfeiler-Lehman kernels [37, 38, 39], and the subgraph pattern based kernels [40, 41]. These two kinds of graph kernels

are both defined based on the subgraph structures. Our SRF kernel differs from them in a number of ways. First, the neighborhood aggregation based graph kernels aggregates the neighborhood's label information while our SGK kernel aggregates the neighborhood information based on the continuous feature vectors. In our semi-supervised setting, where the labeled data is scarce, it is infeasible to straightforwardly apply these graph kernels to our problem. Second, the subgraph pattern based graph kernels proposes to measure the similarities between two graphs by counting the occurrences of subgraph patterns (graphlets) of fixed size. These graph kernels, which are obtained based only on graph structures, do not consider any labels or feature vectors. It is also computationally expensive to find all the subgraph patterns (graphlets) in graphs. By contrast, our SGK kernel measures similarities between nodes by aggregating the $L$-hop neighborhood's feature information using an aggregation function, and it is also more computationally efficient to obtain these subgraph representations.

**Random Fourier Feature.** The random Fourier feature (RFF) [43] is a powerful technique for kernel approximation [44, 45, 46]. It has been successfully applied to address the high computational cost of GPs [43, 47, 48, 45, 49]. Typically, the sparse spectrum GP (SSGP) [47] proposes to directly optimize spectral points along with the hyperparameters. The variational method used in [48] infers the posteriors over the frequencies of the kernel to avoid potential overfitting of SSGP. In comparison, we learn the SRF kernel by inferring the spectral distribution in a data-driven way, which makes the learnt kernel more flexible.

We would like to clarify the differences between RFF and our SRF kernel. The RFF is a technique to approximate a pre-defined kernel function, which requires calculating the spectral distribution of the pre-defined kernel function through the Fourier transform (we refer to [43] or Appendix A for more details about RFF). We emphasize that for most complex kernel functions, it is not feasible to obtain the corresponding spectral distributions due to the intractability of Fourier transform. This is also the reason why RFF is only applicable to a small range of simple kernel functions so far, such as RBF kernel and arc-cosine kernel. By contrast, our SRF kernel is a kernel function directly defined with a spectral distribution (we define SRF kernel with a hierarchy Bayesian distribution in our paper) based on Bochner's theorem. Similar ideas have also been explored to obtain time embedding [50]. The definition of SRF kernel sidesteps calculating the potentially intractable spectral distribution. Also, we can obtain a complex and expressive kernel by defining a complicated spectral distribution. Hence, we generalize RFF to a family of complex and expressive kernel functions, i.e., SRF kernels.

**Graph Neural Networks.** Graph neural networks (GNNs) have been proposed for graph learning. They leverage both graph structures and features to learn useful representations, and have achieved state-of-the-art performance in some graph learning tasks [16, 17, 51, 52, 53, 54, 55]. Compared with GNNs, which are parametric deterministic models, our GPSRF is a Bayesian non-parametric GP model. Moreover, GPSRF enables to incorporate background knowledge by selecting a proper kernel function, and obtain uncertainty estimations of predictions, which are infeasible for GNNs.

## 3 Preliminaries

### 3.1 Gaussian Process with Random Fourier Feature

A Gaussian Process defines a distribution in a function space, which assumes that the marginal distribution over function values of any finite inputs is Gaussian. Specifically, a GP places prior distribution over functions, i.e., $f(\boldsymbol{x}) \sim \mathrm{GP}(\boldsymbol{0}, k_\theta(\boldsymbol{x}, \boldsymbol{x}'))$, where $k_\theta(\boldsymbol{x}, \boldsymbol{x}') \in \mathbb{R}$ denotes the kernel (covariance) function with parameters $\theta$ between $\boldsymbol{x}, \boldsymbol{x}' \in \mathbb{R}^F$. One critical challenge of GPs is the cubic computational cost incurred by the inverse of kernel matrix in posterior inference [10]. To address the challenge, the random Fourier feature (RFF) method has been proposed for scalable learning [43, 45]. Based on the Bochner's theorem [22], the RFF method proposes to approximate shift-invariant kernels with explicit feature mappings [43]. Specifically, for a shift-invariant kernel $k_\theta(\boldsymbol{x}, \boldsymbol{x}') = k_\theta(\boldsymbol{r})$, where $\boldsymbol{r} = \boldsymbol{x} - \boldsymbol{x}'$, the Fourier duality between the kernel and spectral distribution [10] enables to define an explicit feature mapping function:

$$\phi_{\boldsymbol{\omega}}(\boldsymbol{x}) = \sqrt{\frac{2}{M}} \bigg[ \cos(2\pi \boldsymbol{\omega}_m^\top \boldsymbol{x} + b_m) \bigg]_{m=1}^M, \tag{1}$$

such that the inner product between the feature maps is an unbiased estimate of the kernel function, i.e., $k_\theta(\boldsymbol{r}) \approx \phi_{\boldsymbol{\omega}}(\boldsymbol{x})^\top \phi_{\boldsymbol{\omega}}(\boldsymbol{x}')$. In Eq. (1), $\boldsymbol{\omega}_m \in \mathbb{R}^F$ and $b_m$ are samples drawn from the spectral distribution $p(\boldsymbol{\omega})$ and the uniform distribution $\mathrm{Unif}[0, 2\pi]$ respectively, and $M$ represents the total

number of samples. The features obtained with $\phi_{\boldsymbol{\omega}}$ is referred to as the random Fourier features [43]. More details about RFF can be found in Appendix A.

When modeling graph-structured data with GPs, in addition to the inference problem, it is nontrivial to choose a suitable kernel function as: (1) the RFF method is limited to a small range of kernels since it requires to calculate the spectral distribution, which is not always feasible due to the intractability of Fourier transform; (2) a majority of GP kernels are feature-based, which capture similarities based solely on features, making them inefficient in capturing the local geometric structures in graphs.

## 3.2 Graph Neural Network

Recently, graph neural networks [16, 17, 51, 42] have been proposed to effectively leverage geometric structures for graph learning. They aim to learn object representations by capturing the structural information within the neighborhoods of these objects. Specifically, they follow an iterative aggregation (message passing) scheme to update the object representations. For an $L$-layer GNN, the update function at the $l$-th layer is represented as:

$$\mathbf{b}_i^{(l)} = \text{AGGREGTION}(\{\mathbf{h}_j^{(l-1)} : j \in Ne(i)\}), \tag{2}$$

$$\mathbf{h}_i^{(l)} = \text{COMBINE}(\mathbf{h}_i^{(l-1)}, \mathbf{b}_i^{(l)}), \tag{3}$$

where $\mathbf{h}_i^{(l)}$ is the embedding of object $v_i$ at the $l$-th layer with $\mathbf{h}_i^{(0)} = \boldsymbol{x}_i$, $Ne(i)$ is a set of adjacent objects to $v_i$, and $\text{AGGREGATION}(\cdot)$ and $\text{COMBINE}(\cdot)$ are the component functions of GNN. Compared with GPs, GNNs suffer from several drawbacks: it is difficult for GNNs to (1) incorporate background knowledge of the learning problem and (2) estimate uncertainties for predictions.

# 4 Task Formulation of Graph Learning

Graph-structured data are ubiquitous in the real world with a variety of applications, such as object classification [16] and link prediction [56]. In this paper, we instantiate graph learning with the task of object classification in the semi-supervised setting, as many other applications can be reformulated into this task [57]. Let $\mathcal{G} = (\mathcal{V}, \mathcal{E}, \mathbf{X}, \mathbf{Y}^l)$ represent a partially-labeled graph where $\mathcal{V} = \{v_i\}_{i=1}^N$ is a set of objects, $\mathcal{E}$ is a set of edges and $N$ is the total number of objects. The object features are denoted as $\mathbf{X} \in \mathbb{R}^{N \times F}$, where $F$ is the feature dimension. The labels $\mathbf{Y}^l \in \mathbb{R}^{N_l \times C}$ are in the one-hot form, where $N_l$ and $C$ represent the number of labeled objects and the number of classes, respectively. Given the partially-labeled graph, the goal of object classification is to predict the labels of the remaining $N_u$ unlabeled objects, i.e., $\mathbf{Y}^u \in \mathbb{R}^{N_u \times C}$.

# 5 Structure-Aware Random Fourier Kernel

To address the aforementioned issues of GPs on graph-structured data (Section 3.1), we propose a novel Structure-aware Random Fourier (SRF) kernel. We introduce the kernel in this section and detail a GP model with this kernel in next section.

## 5.1 Random Fourier Kernel

The Bochner's theorem [22] states that the Fourier transform of a positive finite measure is a positive definite shift-invariant kernel. Moreover, when the kernel is properly scaled, the measure has a density [43], called the spectral distribution $p(\boldsymbol{\omega})$. Inspired by the Fourier duality between the kernel and the spectral distribution, we propose to define a kernel function with a spectral distribution, which we refer to as the Random Fourier (RF) kernel. The RF kernel is advantageous in that: (1) RF kernel naturally enables a random Fourier feature (RFF) formulation for scalable kernel learning; (2) RF kernel transforms the kernel learning problem to the distribution inference problem, i.e., we can learn an appropriate kernel for data by inferring a proper spectral distribution, and the resulting kernel will be fairly expressive if the dual spectral distribution is adequately complex.

To clarify the first point, it is worth noting that RF kernel is a kernel function defined with spectral distribution while RFF is a method for approximating a kernel function. Compared with other kernels, it is convenient for RF kernel to leverage RFF for scalable learning, as it is unnecessary to compute

the potentially intractable spectral distribution from the kernel function. The second point states that we can obtain an expressive kernel by defining a complicated spectral distribution. For example, the spectral mixture (SM) kernel [2] can be considered as a special case of RF kernel by setting the spectral distribution as mixture of Gaussians. Compared with SM kernel, our RF kernel is more general as the spectral distribution can take on many forms, and the resulting kernel function may not have a closed-form solution, which motivates us to utilize RFF for approximation.

To efficiently infer the spectral distribution, it is convenient to define the spectral distribution as reparameterizable distributions [58], such as Gaussians. However, they may not be rich enough to represent expressive kernels. For examples, the unimodal reparameterizable distributions are difficult to reconstruct the kernels whose spectral distributions are multimodal. To obtain a flexible and expressive kernel for the complicated functions over graph-structured data, we define the spectral distribution in a hierarchical Bayesian manner: $p_\theta(\boldsymbol{\omega}) = \int p(\boldsymbol{\epsilon})p_\theta(\boldsymbol{\omega}|\boldsymbol{\epsilon})d\boldsymbol{\epsilon}$, where $\theta$ is the parameters of the conditional distribution and $p(\boldsymbol{\epsilon})$ is the base distribution. The dependence of conditional distribution $p_\theta(\boldsymbol{\omega}|\boldsymbol{\epsilon})$ on random variable $\boldsymbol{\epsilon}$ can be arbitrarily complex. To enable efficient optimization of $\theta$, we assume that the conditional distribution $p_\theta(\boldsymbol{\omega}|\boldsymbol{\epsilon})$ is reparameterizable (such as Gaussian or exponential distributions). As a result, we can obtain samples from the conditional distribution by drawing an auxiliary variable $\boldsymbol{u}$, and obtain $\boldsymbol{\omega}$ as a deterministic function $h_\theta(\cdot)$ of the sampled $\boldsymbol{u}$,

$$\boldsymbol{\epsilon} \sim p(\boldsymbol{\epsilon}), \boldsymbol{u} \sim p(\boldsymbol{u}), \boldsymbol{\omega} = h_\theta(\boldsymbol{u}; \boldsymbol{\epsilon}). \tag{4}$$

With the spectral points sampled from the spectral distribution, i.e., $\boldsymbol{\omega}$, we can utilize the RFF, which is defined in Eq. (1), to approximate the RF kernel. Consequently, the RF kernel between two objects is given by: $\tilde{k}_\theta(\boldsymbol{x}_i, \boldsymbol{x}_j) = \phi_{\boldsymbol{\omega}}(\boldsymbol{x}_i)^\top \phi_{\boldsymbol{\omega}}(\boldsymbol{x}_j)$. By learning the parameters of the spectral distribution $p_\theta(\boldsymbol{\omega})$ in a data-driven way, we are able to implicitly learn a flexible and expressive kernel function.

## 5.2 Structure-Aware Random Fourier Kernel

Despite the RF kernel is expressive and scalable, it is still feature-based and may be inefficient in capturing the local geometric structures in graphs. To address this issue, we further propose a Structure-aware Random Fourier (SRF) kernel, which is a RF kernel that takes $L$-hop subgraphs centered at the objects as inputs. Specifically, the SRF kernel between two objects is defined as:

$$k_\theta(v_i, v_j) = \tilde{k}_\theta\big(g(\mathcal{G}_i^L), g(\mathcal{G}_j^L)\big) = \phi_{\boldsymbol{\omega}}(g(\mathcal{G}_i^L))^\top \phi_{\boldsymbol{\omega}}(g(\mathcal{G}_j^L)), \tag{5}$$

where $\mathcal{G}_i^L = \big(\boldsymbol{\mathcal{V}}_i, \boldsymbol{\mathcal{E}}_i, \{\boldsymbol{x}_j\}_{j \in \boldsymbol{\mathcal{V}}_i}\big)$ represents the $L$-hop subgraph centered at object $v_i$, with $\boldsymbol{\mathcal{V}}_i$, $\boldsymbol{\mathcal{E}}_i$ being the set of objects and edges, respectively. The subgraph aggregation function $g$ outputs subgraph representations. The SRF kernel is a general framework as we can consider diverse subgraph aggregation functions, such as summation, pooling operation [59] and graph kernels [60]. Inspired by the success of GNN in aggregating the neighborhood information, we set $g$ as an $L$-layer graph neural network (defined in Eq. (3)). We use the node-level representation of node $v_i$ output by GNN as the result of function $g(\mathcal{G}_i^L)$. With the representational power and learning biases of GNNs, $g$ is supposed to effectively capture the local geometric structure of objects.

# 6 Gaussian Process Model with SRF Kernel

After developing a novel SRF kernel, we equip GP with this kernel for graph learning, that is GPSRF (Section 6.1). To efficiently optimize GPSRF, we develop a variational EM algorithm (Section 6.2).

## 6.1 Graph GP Model

We define a vector-valued stochastic function $f : \mathbb{R}^F \to \mathbb{R}^D$ over graph-structured data, where $D$ is the output dimensions. The joint probability of our graph GP model is defined as: $p_\theta(\mathbf{Y}, \mathbf{F}|\mathbf{X}, \boldsymbol{\mathcal{E}}) = p_\theta(\mathbf{F}|\mathbf{X}, \boldsymbol{\mathcal{E}}) \prod_{i=1}^N p_\theta(\mathbf{Y}_i|\mathbf{F}_i)$, where $\mathbf{Y} = [\mathbf{Y}^l; \mathbf{Y}^u]$ and $\mathbf{F} = f(\mathbf{X})$ represent the labels and function values of all the objects, respectively. The likelihood function in our model is defined with the softmax function in the following way: $p_\theta(\mathbf{Y}_i|\mathbf{F}_i) = \frac{\exp_\theta(\mathbf{Y}_i, \mathbf{F}_i)}{\sum_{c=1}^C \exp_\theta(\mathbf{e}_c, \mathbf{F}_i)}$, where $\mathbf{e}_c$ is the indicator vector with the $c$-th element being 1 and the remainder 0. The stochastic function is assumed to follow a GP prior. By using the proposed SRF kernel, this GP prior is given by:

$$p_\theta(\mathbf{F}|\mathbf{X}, \boldsymbol{\mathcal{E}}) = \mathcal{N}(\mathbf{F}; \mathbf{0}, \Phi\Phi^\top), \tag{6}$$

where $\Phi = [\phi_{\boldsymbol{\omega}}(\mathbf{h}_1), \ldots, \phi_{\boldsymbol{\omega}}(\mathbf{h}_N)]^\top$ and $\mathbf{h}_i = g(\mathcal{G}_i^L)$ for all $i = 1, \ldots, N$. It is well-known that GP is equivalent to Bayesian linear model with basis functions $\psi(\mathbf{h}) : \mathcal{X} \to \mathcal{H}$, which transforms the features into a high (possibly infinite) dimensional Hilbert space $\mathcal{H}$ such that the kernel $k_\theta(\mathbf{h}_i, \mathbf{h}_j)$ can be expressed as the inner product of basis functions in Hilbert space [10], i.e., $k_\theta(\mathbf{h}_i, \mathbf{h}_j) = \langle \psi(\mathbf{h}_i), \psi(\mathbf{h}_j) \rangle_{\mathcal{H}}$. We propose to approximate the basis functions $\psi$ with the random Fourier feature maps $\phi_{\boldsymbol{\omega}}$ such that: $\langle \psi(\mathbf{h}_i), \psi(\mathbf{h}_j) \rangle_{\mathcal{H}} \approx \langle \phi_{\boldsymbol{\omega}}(\mathbf{h}_i), \phi_{\boldsymbol{\omega}}(\mathbf{h}_j) \rangle_{\mathbb{R}^M}$. Consequently, the GP-distributed function $f$ can be equivalently represented as: $\mathbf{F}_i = f(\mathbf{h}_i) = \mathbf{A} \, \phi_{\boldsymbol{\omega}}(\mathbf{h}_i)$, where $\mathbf{A} = [\mathbf{a}_1, \ldots, \mathbf{a}_D]^\top$ and $\mathbf{a}_d \sim \mathcal{N}(\mathbf{0}, \mathbf{I}_M)$ for all $d = 1, \ldots, D$. We refer to $\mathbf{A}$ as the random Fourier feature coefficients of the kernel-dependent feature mappings $\phi_{\boldsymbol{\omega}}$.

## 6.2 Variational EM

With the random Fourier feature formulation of our GPSRF, the joint distribution is equivalently represented as: $p_\theta(\mathbf{Y}, \mathbf{A}|\mathbf{X}) = p_\theta(\mathbf{Y}|\mathbf{A}, \mathbf{X}) \, p(\mathbf{A})$. Note that $\mathbf{Y}$ represents the labels of all the objects and we omit the dependence on network structure $\mathcal{E}$ for clarity. To make predictions for the unlabeled objects, we need to infer the posterior $p_\theta(\mathbf{A}, \mathbf{Y}^u|\mathbf{X}, \mathbf{Y}^l)$, which is intractable for our multi-classification model. To address this issue, we develop a variational Expectation-Maximization (EM) algorithm [23, 57]. Specifically, we introduce variational distributions over latent variables $q_\varphi(\mathbf{A}, \mathbf{Y}^u|\mathbf{X}) = q_\varphi(\mathbf{A}) \, p_\theta(\mathbf{Y}^u|\mathbf{A}, \mathbf{X})$. The log-likelihood of observations can be rewritten as:

$$\log p_\theta(\mathbf{Y}^l|\mathbf{X}) = \underbrace{\mathbb{E}_{q_\varphi(\mathbf{A})} \left[ \log p_\theta(\mathbf{Y}^l|\mathbf{A}, \mathbf{X}) + \log \frac{p(\mathbf{A})}{q_\varphi(\mathbf{A})} \right]}_{\mathcal{L}(\theta, \varphi; \mathbf{X}, \mathbf{Y}^l)} + \mathrm{KL}\big(q_\varphi(\mathbf{A}) || p_\theta(\mathbf{A}|\mathbf{X}, \mathbf{Y}^l)\big), \quad (7)$$

where the expectation term, i.e., $\mathcal{L}(\theta, \varphi; \mathbf{X}, \mathbf{Y}^l)$, is the Evidence Lower BOund (ELBO), and the full derivation can be found in Appendix B. According to variational EM, the model parameters, i.e., $\varphi$ and $\theta$, are alternatively learned through an inference and a learning procedure.

**Inference Procedure.** In the inference procedure (E-step), the model estimates true posteriors over latent variables $p_\theta(\mathbf{A}, \mathbf{Y}^u|\mathbf{X}, \mathbf{Y}^l)$. Following the principle of variational EM [23], we fix $p_\theta$ and update the variational distributions to approximate the true posteriors. We further define the variational distributions over latent variables $\mathbf{A}$ as Gaussian distributions, i.e., $q_\varphi(\mathbf{A}) = \prod_{d=1}^D \mathcal{N}(\mathbf{a}_d; \mathbf{m}_d, \mathbf{S}_d)$. Consequently, the variational parameters at step $t$ are optimized as:

$$\varphi^t = \arg\min_\varphi \mathrm{KL}\big(q_\varphi(\mathbf{A}, \mathbf{Y}^u|\mathbf{X}) \, || \, p_{\theta^{t-1}}(\mathbf{A}, \mathbf{Y}^u|\mathbf{X}, \mathbf{Y}^l)\big)$$
$$= \arg\min_\varphi \mathrm{KL}\big(q_\varphi(\mathbf{A}) \, || \, p_{\theta^{t-1}}(\mathbf{A}|\mathbf{X}, \mathbf{Y}^l)\big). \quad (8)$$

According to Eq. (7), minimizing this KL divergence is equivalent to maximizing the ELBO, hence we can optimize the variational parameters by maximizing ELBO using the stochastic back propagation and reparameterization trick [58], where the generative parameters $\theta^{t-1}$ are kept fixed.

**Learning Procedure.** In the learning procedure (M-step), the goal is to update the generative parameters $\theta$. Specifically, we will fix $q_\varphi$ and update $p_\theta$ at step $t$ as:

$$\theta^t = \arg\max_\theta \mathbb{E}_{q_{\varphi^t}(\mathbf{A}, \mathbf{Y}^u|\mathbf{X})} \big[ \log p_\theta(\mathbf{Y}, \mathbf{A}|\mathbf{X}) \big]$$
$$= \arg\max_\theta \mathbb{E}_{q_{\varphi^t}(\mathbf{A}, \mathbf{Y}^u|\mathbf{X})} [\log p_\theta(\mathbf{Y}|\mathbf{A}, \mathbf{X})] + \mathrm{const}. \quad (9)$$

To effectively optimize the parameters, i.e., $\varphi$ and $\theta$, we first pre-train the inference model $q_\varphi$ for a few steps and then perform alternate optimization between an inference procedure and a learning procedure, following the previous work [57]. Also, the subgraph transformation function $g$ is learned in conjunction with model parameters. Such a joint learning can make our model more adaptive to the training data and obtain better performance. After optimization, the labels of the unlabeled objects can be predicted with the generative model $p_\theta$. The pseudo codes can be found in Appendix C.

## 7 Discussions

**Computational Complexity.** The time complexity of GPSRF is $\mathcal{O}(NM(\tilde{F} + D))$, where $M, \tilde{F}, D$ represent the the number of samples from $p(\boldsymbol{\omega})$, the dimension of subgraph representations (i.e., $\mathbf{h}$)

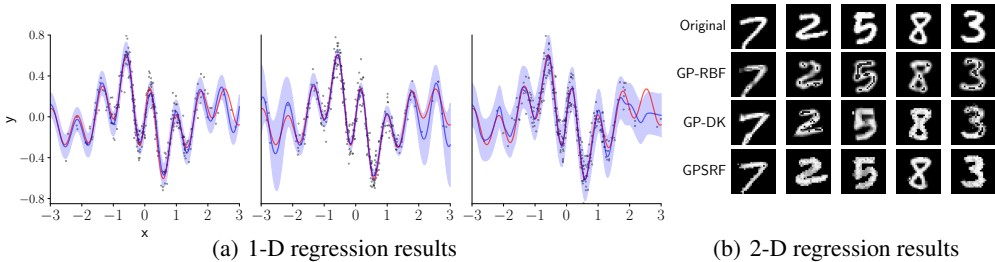

(a) 1-D regression results          (b) 2-D regression results

Figure 1: **(a)** The 1-dimension regression results of the proposed RF kernel $\tilde{k}_\theta$ (**left**) compared with the RBF kernel (**middle**) and deep kernel (**right**). The gray-dot points are the noisy training data, the red lines are the ground-truth, the blue lines are the interpolation results of different kernels, and the shaded areas represent 2-standard deviations. **(b)** Pixel intensity (2-dimension) regression on MNIST. Each row represents the predictions of GP with a specific kernel on different digits.

and the dimension of the output of function $f$, respectively. Compared with the GPs with variational inducing point methods [3] for optimization, which has a time complexity of $\mathcal{O}(ND(P^3 + P^2\tilde{F}))$, where $P$ is the number of inducing points, our method can reduce the time complexity. Since the likelihood function of GPSRF factorizes across observations, we can sample a mini-batch of data for training. In such case, the time complexity can be further reduced to $\mathcal{O}(SM(\tilde{F} + D))$, where $S \ll N$ is mini-batch size. The empirical results in Appendix E.6 also validate that GPSRF has a reduced time complexity.

**Applications.** Despite GPSRF is presented with the instance of semi-supervised object classification task, it can be easily extended to other graph learning tasks by modifying the likelihood function. For example, it can be naturally applied to the link prediction task [61, 62], where the model is given partially observed links and the goal is to predict the remaining links. The link prediction task can also be regarded as a classification task where each link is treated as a binary random variable. Specifically, for an edge $\mathcal{E}_{ij}$ between two objects, we can concatenate the function values of these two objects, i.e., $\mathbf{F}_i$ and $\mathbf{F}_j$, as the function values for the edge. By replacing function values of objects with those of edges in likelihood function, we can obtain GPSRF for link prediction task.

**The pseudo-label perspective of the learning procedure.** In the learning procedure of variational EM algorithm, we need to sample labels of unlabeled objects from the variational distribution $q_\varphi(\mathbf{A}, \mathbf{Y}^u|\mathbf{X})$ for updating the generative parameters $\theta$. If we sample the label of each unlabeled object as the class that has maximum probability in the variational distribution, our method is closely related to the pseudo-label technique [63], which labels each unlabeled object with the class that has the maximum predictive probability and use both the true labels and the pseudo-labels to update the model parameters. Compared with the pseudo-label technique where the pseudo-labels are kept fixed after a pre-training stage, the "pseudo-labels" in our method are dynamically sampled from the variational distribution at each iteration. Moreover, our method only use the "pseudo-labels" in learning procedure (M-step) while in the inference procedure (E-step), only true labels are used. From this perspective, similar to some pseudo-label techniques [64] which only use an unlabeled object for training when its maximum predicted probability is above a certain threshold, we can also introduce a threshold $\tau$ and only include the comparatively accurate "pseudo-labels" for optimization. Empirically, we found GPSRF achieves better performance when introducing the threshold $\tau$ as it can alleviate the overfitting problem in semi-supervised setting (see Section 8.3 for details).

## 8 Experiments

To examine the proposed kernel and GP model, we study the following research questions. (***RQ1***) How does the RF kernel $\tilde{k}_\theta$ (defined in Section 5.1) perform in regression tasks compared with existing kernels? (***RQ2***) Can GPSRF outperform baselines in the graph learning tasks, such as semi-supervised object classification and link prediction? (***RQ3***) How does the number of samples from spectral distribution $M$ affect the performance of GPSRF? (***RQ4***) What are the effects of each component, i.e., subgraph aggregation function $g$ and RF kernel $\tilde{k}_\theta$, in our SRF kernel? In addition to these research questions, further experimental results can be found in Appendix E.

Table 1: Classification accuracies (%) of GPSRF and baselines on different datasets. The best and second best performance under each dataset are marked with boldface and underline, respectively.

| Category | Method | Cora | Citeseer | Pubmed | Photo | Computers |
|---|---|---|---|---|---|---|
| **GP** | GP-RBF | $60.8 \pm 0.3$ | $55.4 \pm 0.3$ | $74.2 \pm 0.5$ | $82.4 \pm 0.2$ | $68.8 \pm 0.5$ |
| | GP-SM | $63.1 \pm 0.3$ | $58.0 \pm 0.4$ | $75.1 \pm 0.4$ | $86.8 \pm 0.3$ | $71.2 \pm 0.6$ |
| | GP-DIF | $68.1 \pm 0.4$ | $47.9 \pm 0.4$ | $65.3 \pm 0.5$ | $77.8 \pm 0.6$ | $63.4 \pm 0.5$ |
| | GP-DK | $72.6 \pm 1.0$ | $60.4 \pm 0.6$ | $76.2 \pm 0.8$ | $90.3 \pm 0.4$ | $74.5 \pm 0.9$ |
| | GGP | $81.1 \pm 0.5$ | $69.3 \pm 0.4$ | $76.7 \pm 0.4$ | $91.2 \pm 0.3$ | $79.7 \pm 0.8$ |
| **GNN** | GraphSAGE | $80.6 \pm 0.3$ | $68.6 \pm 0.8$ | $77.9 \pm 0.2$ | $84.7 \pm 0.7$ | $78.5 \pm 1.1$ |
| | GCN | $81.4 \pm 0.5$ | $70.3 \pm 0.7$ | $78.9 \pm 0.3$ | $88.2 \pm 0.4$ | $83.0 \pm 0.8$ |
| | GAT | $\underline{82.8 \pm 0.7}$ | $\underline{72.1 \pm 0.4}$ | $\underline{79.2 \pm 0.2}$ | $85.3 \pm 0.4$ | $75.6 \pm 0.7$ |
| | DGI | $82.3 \pm 0.6$ | $71.5 \pm 0.7$ | $78.4 \pm 0.5$ | $90.8 \pm 0.9$ | $82.1 \pm 1.2$ |
| **GPSRF** | w/o $\tau$ | $81.9 \pm 0.6$ | $71.4 \pm 0.5$ | $78.7 \pm 0.4$ | $\underline{91.9 \pm 0.4}$ | $\underline{85.8 \pm 0.8}$ |
| | with $\tau$ | $\mathbf{83.6 \pm 0.3}$ | $\mathbf{72.7 \pm 0.5}$ | $\mathbf{79.6 \pm 0.5}$ | $\mathbf{92.2 \pm 0.3}$ | $\mathbf{86.5 \pm 0.2}$ |

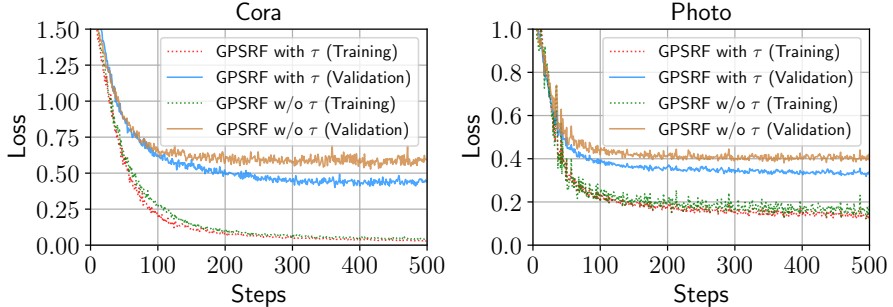

Figure 2: Training and validation loss of GPSRF under different settings on Cora and Photo datasets.

## 8.1 Experimental Settings

To evaluate GPSRF in graph learning tasks, we conduct experiments on three benchmark citation networks [65], including Cora, Citeseer and Pubmed, and two Amazon product co-purchase networks [66], including Photo and Computers. We use the standard split of the citation networks [16]. For the co-purchase networks, we randomly select 20% of nodes as training nodes, 10% of nodes as validation nodes and the remaining nodes are treated as test nodes. In object classification tasks, we take GP models with different kernels as baselines, including the RBF kernel, the SM kernel [2], the Laplacian kernel [19] and the deep kernel [12], which are denoted as GP-RBF, GP-SM, GP-DIF and GP-DK, respectively. Moreover, we also compare against some strong graph learning methods, including GCN [16], GraphSAGE [17], GAT [51], DGI [67] and GGP [18]. For link prediction, we take node2vec [68], GAE/VGAE [61] and GraphSAGE as baselines. Throughout the experiments, to enable mini-batch training, we set the transformation function $g$ as two-layer GraphSAGE network with mean aggregator, where both the aggregation function in Eq. (2) and the combination function in Eq. (3) are mean functions. As for the RF kernel function $\tilde{k}_\theta$, we set the base distribution $p(\epsilon)$ as the standard Gaussian and use the Gaussian conditional $p_\theta(\omega|\epsilon) = \mathcal{N}(\omega; \mu_\theta(\epsilon), \sigma_\theta(\epsilon))$, whose mean and variance are parameterized with neural networks. The parameters of our model are learned with the Adam optimizer [69] through alternating steps of an inference procedure and a learning procedure. More details about experimental settings of our GPSRF and baselines are provided in Appendix D.

## 8.2 Results for Answering *RQ1*: Regression Tasks

To examine the performance of RF kernel $\tilde{k}_\theta$, we compare it with RBF kernel and deep kernel in both 1-D regression and 2-D regression tasks. Specifically, for 1-D regression task, the goal is to fit a target function $y = \frac{\cos 5x * \sin 3x}{|x|+1}$ with some noisy observations. We randomly select 200 data-points in the range of $[-3, 3]$ and add noises to their function values, which are drawn from $\mathcal{N}(0, 0.01)$, to obtain the training set. The regression results are presented in Figure 1(a). It shows that, compared with RBF and deep kernels, the shape of function produced with RF kernel can better match the shape of target function, especially on the region where the training data are sparse. To further examine the expressiveness of RF kernel, we apply it on a more complex 2-D regression task. Following [70], we

Table 2: Link prediction performance (AUC) (%) of GPSRF and baselines on different datasets.

| Method | Cora | Citeseer | Pubmed | Photo | Computers |
|--------|------|----------|--------|-------|-----------|
| **node2vec** | $81.3 \pm 0.3$ | $79.6 \pm 0.5$ | $87.7 \pm 0.1$ | $83.1 \pm 0.4$ | $79.5 \pm 0.7$ |
| **GraphSAGE** | $89.2 \pm 0.3$ | $91.3 \pm 0.2$ | $93.5 \pm 0.2$ | $87.4 \pm 0.7$ | $81.8 \pm 0.6$ |
| **GAE** | $92.3 \pm 0.3$ | $89.4 \pm 0.6$ | $95.9 \pm 0.2$ | $90.1 \pm 0.8$ | $86.4 \pm 1.1$ |
| **VGAE** | $92.2 \pm 0.2$ | $91.5 \pm 1.1$ | $94.4 \pm 1.0$ | $90.8 \pm 0.4$ | $87.1 \pm 0.9$ |
| **GPSRF** | $\mathbf{96.7 \pm 0.3}$ | $\mathbf{97.2 \pm 0.4}$ | $\mathbf{96.5 \pm 0.2}$ | $\mathbf{92.1 \pm 0.6}$ | $\mathbf{90.8 \pm 0.5}$ |

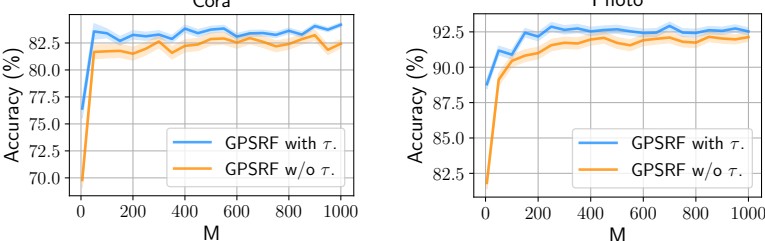

Figure 3: Classification accuracies of GPSRF under different $M$, i.e., the number of samples drawn from the spectral distribution, on Cora and Photo datasets.

treat image completion as a regression task, then given some pixel intensities the goal is to predict the unseen pixel intensities. In this formulation, the input features are the coordinates of each pixel and the function values are the corresponding pixel intensities. We train models on MNIST [71] dataset, where we randomly select 200 pixels of an image as the training set and use the remaining for prediction. The results are presented in Figure 1(b), which shows that the proposed RF kernel can better reconstruct the original images compared with baseline kernels.

## 8.3 Results for Answering *RQ2*: The Performance of GPSRF in Graph Learning Tasks

**Semi-supervised Object Classification.** Next, we examine the performance of GPSRF on semi-supervised object classification task. In addition to the original GPSRF setting, denoted as "GPSRF w/o $\tau$", we further introduce a threshold $\tau$, denoted as "GPSRF with $\tau$", for optimization from the pseudo-label perspective of the learning procedure (see Section 7 for details). In the following, GPSRF refers to "GPSRF w/o $\tau$" unless specifically stated.

The classification results are presented in Table 1. It is shown that GPSRF can outperform all the other GP models except GGP on all the datasets. Compared with GGP, GPSRF performs better on three out of five datasets (Citeseer, Pubmed and Computers) and achieves quite similar results on the other two datasets. These results validate the effectiveness of our SRF kernel for improving the performance of GP on graphs. Compared with parametric GNNs, GPSRF can outperform GraphSAGE and GCN on all the datasets, and achieve comparative results with GAT and DGI. In addition, by introducing the threshold $\tau$, we further improve the performance of GPSRF and obtain the best results on all the datasets. This can be explained from the pseudo-label perspective of the learning procedure: by using the comparatively accurate predictions whose maximum predicted probability is above the threshold for optimization, GPSRF can be prevented from accumulating errors, alleviating the overfitting problem in semi-supervised setting. We also evaluate the performance of GPSRF and baselines under different percentage of labeled data, which is presented in Appendix E.1, and different edge sparsity, which is presented in Appendix E.7.

**Convergence Analysis.** In GPSRF, a variational EM algorithm, which alternates between inference and learning procedure, is used for optimization. We next study the convergence of GPSRF. Specifically, we report the training and validation loss of GPSRF under different settings on Cora and Photo datasets. The results in Figure 2 show that GPSRF can converge within a few hundred steps, which is efficient. Moreover, by introducing the threshold, the validation loss gets smaller, which empirically validates that it can help alleviating the overfitting problem in semi-supervised setting.

**Link Prediction.** Subsequently, we examine the performance of GPSRF in link prediction task. We follow the experimental settings in [61] and use area under the ROC curve (AUC) and average precision (AP) as evaluation metrics. Due to the space limit, we report the AUC results in Table 2 and the AP results in Appendix E.2. We only report results of GPSRF without the threshold $\tau$ as we empirically found similar results when introducing the threshold. Note that the GPs are not taken as

Table 3: Classification accuracies (%) of GPs with different settings of SRF kernel. (Note that "RFF + $g$ + $\tilde{k}_\theta$" is the same as our GPSRF.)

| Method | Cora | Citeseer | Pubmed | Photo | Computers |
|---|---|---|---|---|---|
| **RFF** | $71.4 \pm 1.3$ | $53.8 \pm 0.8$ | $71.7 \pm 0.6$ | $89.5 \pm 0.4$ | $80.4 \pm 0.8$ |
| **RFF** + $g$ | $82.3 \pm 0.5$ | $71.9 \pm 0.6$ | $78.1 \pm 0.5$ | $91.7 \pm 0.3$ | $86.0 \pm 0.5$ |
| **RFF** + $g$ + $\tilde{k}_\theta$ | $83.6 \pm 0.3$ | $72.7 \pm 0.5$ | $79.6 \pm 0.5$ | $92.2 \pm 0.3$ | $86.5 \pm 0.2$ |

baselines since they are tailored for classification tasks. The results show that GPSRF can outperform all the baselines on all datasets. This is because the SRF kernel in GPSRF is structure-aware and can capture the local patterns of the objects, which is beneficial to the success of link prediction.

### 8.4 Results for Answering *RQ3*: The Effect of the Number of Samples

In our GPSRF, we leverage RFF method to approximate the SRF kernel, which requires to sample from the spectral distribution. To understand the effect of the sampling size $M$ on the performance of GPSRF, we further conduct experiments on the object classification task. Specifically, we report the performance of GPSRF under different sampling sizes on Cora and Photo datasets. The experimental results in Figure 3 show that GPSRF can achieve decent results even with a low sampling rate and the performance converges when the sampling size is cross a certain value. These findings show another merit of our GPSRF: it is sufficient for GPSRF to use only a small number of samples for optimization to obtain satisfactory performance, which is efficient.

### 8.5 Results for Answering *RQ4*: Ablation Study

Finally, we conduct ablation study to examine the effect of each component in SRF kernel, i.e., $g$ and $\tilde{k}_\theta$. The experimental results on object classification are presented in Table 3, where "RFF" denotes the random Fourier feature with spectral distribution corresponding to the RBF kernel. It's shown that the performance of RFF can be greatly improved by introducing the function $g$, which is because the $g$ can summarize the local patterns of objects and produce useful representations for them. This result indicates the necessity of considering geometric structures in the kernel when modeling graphs. Moreover, by additionally introducing the RF kernel $\tilde{k}_\theta$, the performance can be further improved as $\tilde{k}_\theta$ is more expressive and flexible than RBF kernel, and is adaptively learned from the data. These results prove the effectiveness of employing both $g$ and $\tilde{k}_\theta$ in the proposed SRF kernel.

## 9 Conclusion

In this paper, to improve GP's performance on graph-structured data, we propose a novel SRF kernel. The SRF kernel is defined with a spectral distribution based on the Fourier duality given by the Bochner's theorem. Compared with feature-based kernels, the SRF kernel enables to leverage both objects features and geometric structures. Based on the SRF kernel, we propose a GP model, GPSRF, for graph learning tasks. To effectively optimize GPSRF, we develop a variational EM algorithm. Empirically, we conduct experiments on five real-world datasets in two graph learning settings, i.e., semi-supervised object classification and link prediction. Experimental results show that our method yields best performance compared with state-of-the-art baselines.

This paper defines SRF kernel with a hierarchy Bayesian spectral distribution and we empirically find this can achieve satisfactory results. However, the performance of many other complicated distributions, such as mixture distributions and flow-based distributions [72], remains unexplored. Moreover, our GPSRF aims to solve learning tasks in homogeneous graphs and it is unclear how it performs on heterogeneous graphs with multiple types of relations and objects. We leave these limitations to further work.

The proposed method can be applied in various graph-based applications and improve the capacity of the current graph learning algorithms. However, when applied practically, it may incur some negative societal impacts. For instance, our method requires offline training, which may results in biased predictions when employed in online scenarios, where the real-time updates are essential. Moreover, the robustness of our method could be further improved to prevent attacks from an adversary, who tries to mislead the algorithm to make false predictions.

## Acknowledgments

This research was supported by the National Natural Science Foundation of China (Grant No. 61906219).

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
