# Appendix: Structure-Aware Random Fourier Kernel for Graphs

**Jinyuan Fang**[1,2]**, Qiang Zhang**[3,4,5]**, Zaiqiao Meng**[6,7]**, Shangsong Liang**[1,2,7*]
[1] School of Computer Science and Engineering, Sun Yat-sen University, China
[2] Guangdong Key Laboratory of Big Data Analysis and Processing, Guangzhou, China
[3] Hangzhou Innovation Center, Zhejiang University, China
[4] College of Computer Science and Technology, Zhejiang University, China
[5] AZFT Knowledge Engine Lab, China
[6] School of Computing Science, University of Glasgow, United Kingdom
[7] Mohamed bin Zayed University of Artificial Intelligence, United Arab Emirates
{fangjy6@gmail.com; qiang.zhang.cs@zju.edu.cn; zaiqiao.meng@gmail.com}
{liangshangsong@gmail.com}

## A    Random Fourier Feature

The random Fourier feature (RFF) is a powerful technique for approximating a shift-invariant kernel function. It is based on the Bochner's theorem [1]:

**Theorem A.1 (Bochner's theorem)** *A continuous, real valued, symmetric and shift-invariant functions $k(\boldsymbol{x}, \boldsymbol{x}') = k(\boldsymbol{r})$ on $\mathbb{R}^F$, where $\boldsymbol{r} = \boldsymbol{x} - \boldsymbol{x}'$, is a positive definite kernel if and only if it is the Fourier transform of a positive finite measure.*

It has been shown in [2] that when the shift-invariant kernel $k(\boldsymbol{r})$ is properly scaled, the measure has a density, called the *spectral distribution $p(\boldsymbol{\omega})$*. This gives rise to the Fourier duality of the kernel function and the spectral distribution, which is known as the Wiener-Khintchin theorem [3]:

$$p(\boldsymbol{\omega}) = \int k(\boldsymbol{r}) e^{-2\pi i \boldsymbol{\omega}^\top \boldsymbol{r}} d\boldsymbol{r}, \tag{10}$$

$$k(\boldsymbol{r}) = \int p(\boldsymbol{\omega}) e^{2\pi i \boldsymbol{\omega}^\top \boldsymbol{r}} d\boldsymbol{\omega}. \tag{11}$$

By using Euler's formula $e^{ix} = \cos x + i \sin x$, the kernel function $k(\boldsymbol{r})$ in Equation 11 can be equivalently represented as:

$$\begin{aligned}
k(\boldsymbol{r}) &= \int p(\boldsymbol{\omega}) e^{2\pi i \boldsymbol{\omega}^\top \boldsymbol{r}} d\boldsymbol{\omega} \\
&= \int p(\boldsymbol{\omega}) \big( \cos(2\pi \boldsymbol{\omega}^\top \boldsymbol{r}) + i \sin(2\pi \boldsymbol{\omega}^\top \boldsymbol{r}) \big) d\boldsymbol{\omega} \\
&= \int p(\boldsymbol{\omega}) \cos(2\pi \boldsymbol{\omega}^\top \boldsymbol{r}) d\boldsymbol{\omega} \\
&= \mathbb{E}_{p(\boldsymbol{\omega})}[\cos(2\pi \boldsymbol{\omega}^\top (\mathbf{x} - \mathbf{x}'))].
\end{aligned} \tag{12}$$

By using the formula proven in [4], which is represented as:

$$\cos(x - y) = \int_0^{2\pi} \frac{1}{2\pi} \sqrt{2} \cos(x + b) \cdot \sqrt{2} \cos(y + b) db, \tag{13}$$

---

*Corresponding Author.

35th Conference on Neural Information Processing Systems (NeurIPS 2021).

we can reformulate Equation 12 as:

$$\cos(2\pi\boldsymbol{\omega}^\top(\mathbf{x}-\mathbf{x}')) = \int_0^{2\pi} \frac{1}{2\pi}\sqrt{2}\cos(2\pi\boldsymbol{\omega}^\top\mathbf{x}+b)\cdot\sqrt{2}\cos(2\pi\boldsymbol{\omega}^\top\mathbf{x}'+b)\,db. \qquad (14)$$

The integral over the random variable $b$ can also be approximated with Monte Carlo method by drawing samples from the uniform distribution $b \sim \mathrm{Unif}[0, 2\pi]$. By drawing $M$ samples $b$ and $\boldsymbol{\omega}$ from the uniform distribution and the spectral distribution, respectively, we can obtain the approximation to the kernel function, which is computed as:

$$k(\boldsymbol{r}) \approx \frac{1}{M}\sum_{m=1}^{M}\sqrt{2}\cos(2\pi\boldsymbol{\omega}_m^\top\mathbf{x}+b_m)\cdot\sqrt{2}\cos(2\pi\boldsymbol{\omega}_m^\top\mathbf{x}'+b_m). \qquad (15)$$

This formulation allows us to define an explicit feature mapping function:

$$\phi_{\boldsymbol{\omega}}(\boldsymbol{x}) = \sqrt{\frac{2}{M}}\left[\cos(2\pi\boldsymbol{\omega}_m^\top\boldsymbol{x}+b_m)\right]_{m=1}^M, \qquad (16)$$

such that the inner product between the feature map is an unbiased estimate of the kernel function, i.e., $k(\boldsymbol{r}) \approx \phi_{\boldsymbol{\omega}}(\boldsymbol{x})^\top\phi_{\boldsymbol{\omega}}(\boldsymbol{x}')$. The features obtained with $\phi_{\boldsymbol{\omega}}$ is referred to as the random Fourier features (RFFs) [2].

In conclusion, to approximate a shift-invariant kernel function, we need to calculate the corresponding spectral distribution according to Equation 10, from which we sample the spectral points, i.e., $\boldsymbol{\omega}_m$, to obtain the random Fourier features. As a result, the kernel function between two data points can be approximated by the inner product of their random Fourier features. However, in most cases, it is not feasible to obtain the spectral distribution of a kernel, due to the intractability of Fourier transform. Hence the RFF is limited to a small range of simple kernel functions such as RBF kernel.

## B  Derivation of Log-likelihood of Observations

By using the proposed Structure-aware Random Fourier (SRF) kernel in the GP model, the joint probability distribution of our model is denoted as:

$$p_\theta(\mathbf{Y}^l, \mathbf{Y}^u, \mathbf{A}|\mathbf{X}) = p_\theta(\mathbf{Y}^l|\mathbf{A}, \mathbf{X})p_\theta(\mathbf{Y}^u|\mathbf{A}, \mathbf{X})p(\mathbf{A}), \qquad (17)$$

where $\mathbf{Y}^l$ and $\mathbf{Y}^u$ represent the one-hot encoding label matrices of labeled objects and unlabeled objects, respectively. Note that here we omit the dependency on the network structures $\mathcal{E}$ for clarity. Following the principle of variational inference, we assume the variational distribution over latent variables, i.e., $\mathbf{A}$ and $\mathbf{Y}^u$, as:

$$q_\varphi(\mathbf{A}, \mathbf{Y}^u|\mathbf{X}) = q_\varphi(\mathbf{A})p_\theta(\mathbf{Y}^u|\mathbf{A}, \mathbf{X}). \qquad (18)$$

As a result, the log-likelihood of the observations can be reformulated as:

$$
\begin{aligned}
\log p_\theta(\mathbf{Y}^l|\mathbf{X}) &= \log p_\theta(\mathbf{Y}^l, \mathbf{Y}^u, \mathbf{A}|\mathbf{X}) - \log p_\theta(\mathbf{Y}^u, \mathbf{A}|\mathbf{X}, \mathbf{Y}^l) \\
&= \log p_\theta(\mathbf{Y}^l, \mathbf{Y}^u, \mathbf{A}|\mathbf{X}) - \log p_\theta(\mathbf{A}|\mathbf{X}, \mathbf{Y}^l) \\
&\quad - \log p_\theta(\mathbf{Y}^u|\mathbf{X}, \mathbf{A}) + \log q_\varphi(\mathbf{A}, \mathbf{Y}^u|\mathbf{X}) \\
&\quad - \log q_\varphi(\mathbf{A}, \mathbf{Y}^u|\mathbf{X}) \\
&= \mathbb{E}_{q_\varphi(\mathbf{A}, \mathbf{Y}^u|\mathbf{X})}\big[\log p_\theta(\mathbf{Y}^l, \mathbf{Y}^u, \mathbf{A}|\mathbf{X}) - \log q_\varphi(\mathbf{A}, \mathbf{Y}^u|\mathbf{X})\big] \\
&\quad + \mathbb{E}_{q_\varphi(\mathbf{A})}\big[\log q_\varphi(\mathbf{A}) - \log p_\theta(\mathbf{A}|\mathbf{X}, \mathbf{Y}^l)\big] \\
&= \mathbb{E}_{q_\varphi(\mathbf{A})}\big[\log p_\theta(\mathbf{Y}^l|\mathbf{A}, \mathbf{X}) + \log p(\mathbf{A}) - \log q_\varphi(\mathbf{A})\big] \\
&\quad + \mathrm{KL}\big(q_\varphi(\mathbf{A})||p_\theta(\mathbf{A}|\mathbf{X}, \mathbf{Y}^l)\big), \qquad (19)
\end{aligned}
$$

where the first expectation term is the Evidence Lower BOund (ELBO) while the second term is the KL divergence between the variational distribution and the true posterior over latent variable $\mathbf{A}$.

---

**Algorithm 1** The proposed GPSRF approach for semi-supervised object classification task.

---

1: **Input:** A partially labeled graph $\mathcal{G} = (\mathcal{V}, \mathcal{E}, \mathbf{X}, \mathbf{Y}^l)$.
2: **Output:** Object labels $\mathbf{Y}^u$ for the unlabeled objects.
3: Pre-train $q_\varphi$ and $g$ according to Eq. (8).
4: **while** *not converge* **do**
5:    ⊡ **E-step: Inference Procedure**
6:    Calculate the subgraph representations $\phi_\omega(g(\mathcal{G}_i^L))$ for all labeled objects, i.e., $i = 1, \ldots N_l$.
7:    Update $q_\varphi$ and $g$ with $\mathbf{Y}^l$ based on Eq. (8).
8:    ⊡ **M-step: Learning Procedure**
9:    Calculate the subgraph representations $\phi_\omega(g(\mathcal{G}_i^L))$ for all the objects, i.e., $i = 1, \ldots N$.
10:    Annotate unlabeled objects with $p_\theta$.
11:    Denote sampled labels for unlabeled objects as $\hat{\mathbf{Y}}^u$ and set $\mathbf{Y} = (\mathbf{Y}^l, \hat{\mathbf{Y}}^u)$.
12:    Update $p_\theta$ and $g$ with $\mathbf{Y}$ based on Eq. (9).
13: **end while**
14: Classify each unlabeled object with $p_\theta$.

---

## C Algorithm Overview

We provide an overview of the optimization process of GPSRF for object classification in Algorithm 1. When training GPSRF, we firstly pre-train the inference model $q_\varphi$ for a few steps and then perform alternate optimization between an inference procedure and a learning procedure. In the inference procedure (E-step), GPSRF calculates representations for the labeled objects and maximizes the ELBO defined in Eq. (7). In the learning procedure (M-step), GPSRF calculates representations for all the objects, annotates the unlabeled objects with labels $\mathbf{Y}^u$ drawn from the variational distribution, and then maximizes the expected log-likelihood in Eq. (9). From the psudo-label perspective of the learning procedure, we can also introduce a threshold $\tau$ and annotate the unlabeled objects with the labels $\tilde{\mathbf{Y}}^u$, whose predicted probability is cross the threshold, for optimization in the learning procedure.

## D Implementation Details

### D.1 Details of the Datasets

We conduct experiments on the following five real-world network datasets, which consists of three benchmark citation networks and two Amazon Co-purchase networks, statistical information of which is provided in Table 1.

- **Cora, Citeseer, Pubmed** [5]: The three datasets are citation networks where the nodes represent the publications and the edges represent the citation links between publications. The features of each node are bag-of-word representations of the corresponding publications. Following the previous work [6], we use the standard train/val/test splits of these datasets in our experiments, where 20 nodes from each class are treated as labeled nodes while the remaining nodes are treated as unlabeled ones.

- **Photo, Computers** [7]: The two datasets are the Amazon product co-purchase networks where the nodes represent the products and the edges between two nodes indicate that two products are frequently bought together. The features of each node are the bag-of-word product reviews and the labels are given by the product category [2]. We randomly select $20\%$ of nodes as training nodes, $10\%$ of nodes as validation nodes and the remaining nodes are treated as test nodes.

Note that the datasets we are using are publicly available network data with their licenses, which have been frequently used in the research community, and they do not contain any personally identifiable information or offensive content.

---

[2] The datasets are available from `https://github.com/rusty1s/pytorch_geometric/tree/master/torch_geometric/datasets`

Table 1: Statistics of the datasets used in the experiments.

| Dataset | Type | #Nodes | #Edges | #Features | #Labels | Label Rate |
|---------|------|--------|--------|-----------|---------|------------|
| Cora | Citation | 2,708 | 5,429 | 1,433 | 7 | 0.052 |
| Citeseer | Citation | 3,312 | 4,660 | 3,703 | 6 | 0.036 |
| Pubmed | Citation | 19,717 | 44338 | 500 | 3 | 0.003 |
| Photo | Co-purchase | 7,650 | 143,663 | 745 | 8 | 0.2 |
| Computers | Co-purchase | 18,333 | 81,894 | 6,805 | 15 | 0.2 |

## D.2 Implementation Details of Baselines

**GP-RBF.** For GP model with the RBF kernel, we use a zero mean function and a softmax-likelihood function, which is the same as our GPSRF. We use the variational inducing point method to train the model [8], where we introduce a set of inducing points and inducing variables for optimization. We implement the model with the GPflow [9] package. In this model, we set the number of functions the same as the number of classes, i.e., $D = C$. We use 500 inducing points and inducing variables, where the variational distribution of inducing variables are chosen to be multivariate Gaussian distribution. The model is trained with Adam optimizer with a learning rate as 0.0005 for 2000 iterations.

**GP-SM.** For GP model with spectral kernel [10], we use the same experimental settings as GP-RBF, where we simply replace the RBF kernel with the spectral kernel function. The spectral kernel function in our experiments is derived by setting the spectral distribution as 3-mixture of Gaussians.

**GP-DK.** The experimental settings are the same as those for GP-RBF and GP-SM. To define the deep kernel, we use a two-layer feed-forward neural network, with 32 Relu units in each hidden layers, to transform the input features and then use the RBF kernel function to calculate the covariance between data points.

**GP-DIF.** In our experiments, we use the diffusion kernel [11], which belongs to the family of structure-based Laplacian kernels. The diffusion kernel are obtained by applying exponential function to the eigenvalues of the graph Laplacian matrix.

**GraphSAGE** [12]. For fair comparison with our GPSRF, the network structures and the parameter settings of GraphSAGE is the same as GPSRF (see Section D.4 and Section D.5 for details).

For other baselines: GCN [6], GAT [13], DGI [14] and GPP [15], we implement the models with the code released by the authors. We tune the hyperparameters of baselines in each task and dataset to be optimal based on their performance on the validation set.

## D.3 Regression Tasks

**1-D regression.** In 1-dimension function regression task, to define the RF kernel $\tilde{k}_\theta$, we choose a 1-dimensional standard Gaussian as the base distribution $p(\epsilon)$ and use the Gaussian conditional $p_\theta(\omega|\epsilon) = \mathcal{N}(\omega|\mu_\theta(\epsilon), \mathrm{diag}(\sigma))$, whose mean is parameterized with a two-layer feed-forward neural network with 10 relu units in each layer. The number of spectral points drawn from the spectral distribution, i.e., $M$, is 16. The Adam optimizer is used for training with a learning rate of 0.001 and the weight decay of 0.0005. We pre-train the inference model $q_\varphi$ for 200 steps and then perform alternate optimization for 2000 steps.

**2-D Regression.** The experimental settings of SRF kernel for the image completion task is similar to the 1-D regression case. However, we set the number of hidden units of neural network as 64 and set the number of samples $M$ as 25.

## D.4 Semi-supervised Object Classification

In GPSRF, for semi-supervised object classification task, we use 2-hop subgraphs centered at the objects as inputs. As a result, we set the transformation function $g$ for producing the representations of subgraphs as a two-layer GraphSAGE network with mean aggregator to enable efficient mini-batch training. In the GraphSAGE network, we use 128 relu units in the first layer and 64 relu

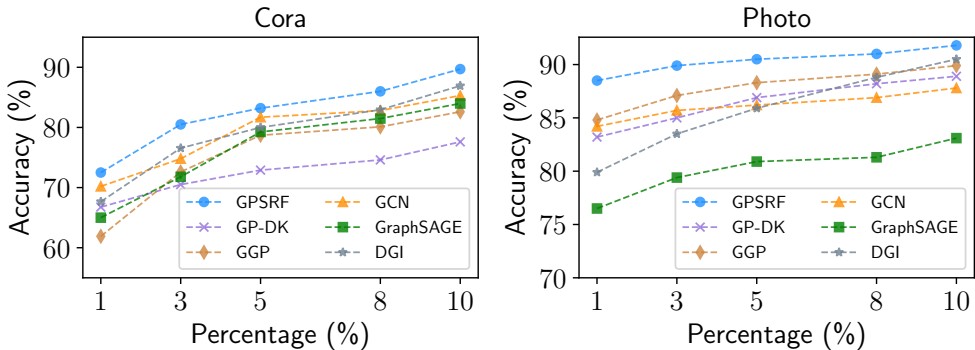

Figure 1: Classification accuracies of GPSRF and baselines on Cora and Photo datasets under different percentage of labeled data.

Table 2: Link prediction performance (AP) (%) of GPSRF and baselines on different datasets.

| Method | Cora | Citeseer | Pubmed | Photo | Computers |
|---|---|---|---|---|---|
| **node2vec** | $83.2 \pm 0.2$ | $82.5 \pm 0.3$ | $85.9 \pm 0.4$ | $85.0 \pm 0.7$ | $81.7 \pm 1.0$ |
| **GraphSAGE** | $89.1 \pm 0.6$ | $92.3 \pm 0.2$ | $95.6 \pm 0.3$ | $86.9 \pm 0.5$ | $82.2 \pm 0.7$ |
| **GAE** | $92.4 \pm 0.1$ | $89.5 \pm 0.4$ | $96.0 \pm 0.3$ | $89.6 \pm 0.4$ | $86.9 \pm 0.5$ |
| **VGAE** | $92.3 \pm 0.3$ | $91.7 \pm 0.9$ | $94.7 \pm 1.0$ | $90.0 \pm 0.6$ | $87.5 \pm 1.2$ |
| **GPSRF** | $\mathbf{97.8 \pm 0.3}$ | $\mathbf{97.5 \pm 0.4}$ | $\mathbf{96.6 \pm 0.2}$ | $\mathbf{93.2 \pm 0.4}$ | $\mathbf{91.3 \pm 0.4}$ |

units in the second layer. For the RF kernel $\tilde{k}_\theta$, we set $p(\epsilon)$ as 1-dimensional standard Gaussian distribution and the Gaussian conditional $p_\theta(\omega|\epsilon)$ is parameterized with two-layer feed-forward neural network with 64 relu units in each hidden layer. To enable efficient training, we assume the variational distribution of each random variable $\mathbf{a}_d$ to have a diagonal covariance structure, i.e., $q_\varphi(\mathbf{A}) = \prod_{d=1}^{D} \prod_{m=1}^{M} \mathcal{N}(m_{d,m}, \sigma_{d,m})$. We set the number of functions to be equal to the number of classes, i.e., $D = C$, on each dataset. Consequently, we can define the likelihood function with the inner product between labels and the function values, i.e., $\exp_\theta(\mathbf{Y}_i, \mathbf{F}_i) = \exp(\mathbf{Y}_i^\top \mathbf{F}_i)$. We ran object classification experiments on a single machine with 8 NVIDIA GeForce RTX 2080Ti with 11GB memory, 56 Intel Xeon CPUs (E5-2680 v4 @ 2.40GHz).

Throughout the semi-supervised object classification tasks, we set the number of samples from the spectral distribution $M$ as 1000. Our model is first pre-trained with the inference procedure for 100 steps and then alternatively optimized for 1000 steps between an inference step and a learning step with a batch size of 512. We use a dropout rate of 0.5 and weight_decay of 0.0005 to alleviate the overfitting problem. The learning rate and the threshold $\tau$ are chosen from $\{0.0005, 0.0015, 0.0020, 0.0025, 0.0030, 0.0040, 0.0050\}$ and $[0.1, 0.9]$, respectively, based on their performance on the validation set. The optimal settings on each dataset are: Cora (E-step learning rate: 0.0005, M-step learning rate: 0.0015, $\tau$: 0.8); Citeseer (E-step learning rate: 0.0005, M-step learning rate: 0.001, $\tau$: 0.8); Pubmed (E-step learning rate: 0.0005, M-step learning rate: 0.0025, $\tau$: 0.9); Photo (E-step learning rate: 0.0005, M-step learning rate: 0.002, $\tau$: 0.7); Computers (E-step learning rate: 0.0005, M-step learning rate: 0.0015, $\tau$: 0.9). In the learning procedure of our variational EM algorithm, we need to sample the labels of unlabeled objects $\mathbf{Y}^u$ from the variational distribution. In our experiments, motivated by the pseudo-label method, we set the labels of unlabeled objects as the classes that have the maximum predicted probability in the variational distribution, which we empirically found can achieve the best performance. We run experiments on each dataset for 10 times and report the average results and standard deviations in the paper.

### D.5 Link Prediction

In the link prediction task, following the experimental settings in [16], we randomly divide edges in the graph into training set (85%), validation set (5%) and test set (10%), and sample an equal number of non-existing edges as the negative samples in all these sets, respectively. To evaluate the link prediction performance, we use the area under the ROC curve (AUC) and average precision (AP) as

evaluation metrics, which are common metrics for evaluating the performance of link prediction in the literature [16].

In GPSRF, link prediction is treated as a binary classification task, i.e., $\mathcal{E}_{ij} \in \{0,1\}$, where we concatenate the function values of two objects as the function values for the edges between them. We also use the two-layer GraphSAGE with mean aggregator as the transformation function. However, we set the number of hidden units in the first and second layer as $64$ and $32$, respectively. The settings of the RF kernel $\tilde{k}_\theta$ is the same as object classification task (see section D.4 for details). We further set the number of functions $D$ as $32$. Hence, for an edge $\mathcal{E}_{ij}$, the likelihood function is defined as:

$$p_\theta(\mathcal{E}_{ij} = 1) = \text{sigmoid}\big(\boldsymbol{r}^\top \text{Concat}(\mathbf{F}_i, \mathbf{F}_j)\big),$$

where $\boldsymbol{r}$ is the weight vector of the linear transformation, $\text{Concat}(\cdot, \cdot)$ represents the concatenation operator between two vectors and $\text{sigmoid}(\cdot)$ is the sigmoid function for producing the likelihood parameter. The optimization of the GP model for link prediction can also be carried out with the variational EM algorithm in a similar manner. We ran link prediction experiments on a single machine with 8 NVIDIA GeForce RTX 2080Ti with 11GB memory, 56 Intel Xeon CPUs (E5-2680 v4 @ 2.40GHz).

In all the datasets, the number of samples $M$ is set to be $256$. For optimization, we use a dropout rate as $0.3$, weight decay as $0.0005$ and a learning rate as $0.005$. Our GPSRF is also pre-trained for 200 steps and alternatively optimized for 2000 steps with a batch size of $512$. Similarly, we also run experiments on each dataset for 10 times and report the average results and standard deviations.

# E   Further Experimental Results

Due to the space limit, we report some additional experimental results in this section. These results include: (1) We report the classification results of GPSRF and baselines under different percentage of labeled data in Section E.1. (2) We report the average precision (AP) results of GPSRF and baselines in link prediction tasks in Section E.2. (3) We examine the effects of the subgraph size $L$ in Section E.3. (4) We examine the effectiveness of the hierarchy Bayesian spectral distribution in the SRF kernel in Section E.4. (5) We examine the effectiveness of the proposed variational EM algorithm in Section E.5. (6) We show that, compared with GPs using variational inducing point method for optimization, GPSRF has a reduced time complexity in Section E.6. (7) Finally, we examine the effects of edge sparsity on the performance of GPSRF in Section E.7.

## E.1   Classification Results under Different Percentage of Labeled Data

To further validate the effectiveness of GPSRF in semi-supervised learning, we evaluate the performance of GPSRF under different percentage of labeled data. Specifically, we vary the label ratio from 1% to 10% on both Cora and Photo datasets, and report the classification results in Figure 1. Note that the results of our GPSRF is obtained after introducing the threshold $\tau$, as we empirically find that this can achieve better performance. The experimental results show that GPSRF always obtain the best performance under different label ratio on both Cora and Photo datasets, which indicates that GPSRF is effective on object classification task even when the labeled data are scarce.

## E.2   Average Precision (AP) Results in Link Prediction

In link prediction task, we take AUC and AP as our evaluation metrics (see D.5 for details). Due to the space limit, we only show the experimental results of AUC in Table 2 of the main text. We further present the experimental results of AP in Table 2. As shown in the table, our GPSRF can also outperform all the other baselines on all the datasets in terms of the AP metrics, which verify the effectiveness of GPSRF in link prediction tasks.

## E.3   The Effect of the Subgraph Size $L$

The proposed SRF kernel takes the $L$-hop subgraphs centered at the objects as input. Consequently, we study how does the size of the subgraphs $L$ affect the performance of GPSRF. Specifically, we vary $L$ from 0 to 3 and report the performance of GPSRF under different settings on each dataset. The experimental results are presented in Table 3. The setting of $L = 0$ represents only the object

Table 3: Classification accuracies (%) of GPSRF with different subgraph size $L$ on each datasets.

| Subgraph Size | Cora | | Citeseer | | Pubmed | | Photo | | Computers | |
|---|---|---|---|---|---|---|---|---|---|---|
| | w/o $\tau$ | with $\tau$ | w/o $\tau$ | with $\tau$ | w/o $\tau$ | with $\tau$ | w/o $\tau$ | with $\tau$ | w/o $\tau$ | with $\tau$ |
| $L = 0$ | 41.8 | 69.9 | 45.1 | 56.4 | 66.7 | 70.4 | 81.0 | 86.2 | 79.6 | 80.7 |
| $L = 1$ | 79.6 | 80.3 | 68.8 | 70.5 | 71.8 | 72.7 | 91.5 | 91.8 | 82.9 | 84.9 |
| $L = 2$ | 81.9 | 83.6 | 71.4 | 72.7 | 78.7 | 79.8 | 91.9 | 92.2 | 85.8 | 86.5 |
| $L = 3$ | 81.0 | 81.3 | - | - | 75.3 | 77.2 | 90.7 | 91.0 | 80.7 | 85.7 |

Table 4: Classification accuracies (%) of GPSRF under different types of spectral distribution.

| Method | Cora | Citeseer | Pubmed | Photo | Computers |
|---|---|---|---|---|---|
| **Gaussian** | $82.5 \pm 0.2$ | $71.8 \pm 0.3$ | $79.7 \pm 0.3$ | $90.5 \pm 0.3$ | $84.6 \pm 0.4$ |
| **Hierarchy** | $\mathbf{83.6 \pm 0.3}$ | $\mathbf{72.7 \pm 0.5}$ | $\mathbf{79.6 \pm 0.5}$ | $\mathbf{92.2 \pm 0.3}$ | $\mathbf{86.5 \pm 0.2}$ |

features are taken as input. In such case, we replace the graph convolution network with two-layer feed-forward neural networks. The results of GPSRF with $L = 3$ on Citesser dataset are missing because the out of memory issue.

Compared to the GPSRF with feed-forward neural networks, GPSRFs with graph neural networks of different sizes can achieve much better performance, which verify the effectiveness of incorporating structural information in the kernel function to improve the performance of GPs when modeling graph data. Moreover, as we increase $L$ from 1 to 2, the performance of GPSRF on each dataset can be improved, which shows that it's more effective to use 2-hop neighbor information than the 1-hop neighbor information. However, as we increase $L$ as 3, the performance of GPSRF begins to decrease, which is in line with the experimental results in [6]. This may be because GPSRF suffers from the overfitting problem when $L$ is greater than 2 since the annotated data are scarce.

### E.4    The Effect of the Hierarchy Bayesian Spectral Distribution

In the proposed SRF kernel function, we use the hierarchy Bayesian spectral distribution $p_\theta(\boldsymbol{\omega}) = \int p(\boldsymbol{\epsilon}) p_\theta(\boldsymbol{\omega}|\boldsymbol{\epsilon}) d\boldsymbol{\epsilon}$ to define an expressive kernel function. To empirically show that the this distribution can define an expressive kernel function, we compare it with Gaussian distribution. The results are presented in Table 4, which shows that the kernel function defined with hierarchy Bayesian distribution can consistently outperform the kernel function defined with the Gaussian distribution. This result verifies that using the hierarchy Bayesian distribution can define a more expressive and flexible kernel function.

### E.5    The Effect of the Variational EM Algorithm

In our GPSRF, we use the proposed SRF kernel to capture the statistical structure and variational EM algorithm for optimization. To examine the effect of variational EM algorithm, we compare it against an end-to-end variational learning method. Specifically, following the variational learning procedure in [17], we introduce a variational posterior over random variables $\mathbf{A}$, i.e., $q_\varphi(\mathbf{A})$, and jointly optimize the variational parameters and generative parameters by maximizing the ELBO in an end-to-end fashion. The experimental results are presented in Table 5, from which we make the following observations: (1) Compared with other GP models (the results are shown in the Table. 1 of the main text), our GPSRF trained in an end-to-end fashion can also achieve better performance, which can be attributed to the effectiveness of the proposed SRF kernel function. (2) Our GPSRF trained with variational EM method can outperform the GPSRF trained with the end-to-end variational learning method. This is because in the learning procedure of the variational EM algorithm, we need to sample the labels of unlabeled objects and use them for optimization. From the psudo-label perspective of this procedure, this is equivalent to augmenting the labels of data, which is helpful to alleviate the overfitting problem in the semi-supervised learning setting. The experimental results demonstrate the effectiveness of the proposed variational EM algorithm.

Table 5: Classification accuracies (%) of GPSRF under different types of variational learning methods. The "**Joint**" refers to the end-to-end variational learning method while "**EM**" refers to the proposed variational EM algorithm.

| Method | Cora | Citeseer | Pubmed | Photo | Computers |
|--------|------|----------|--------|-------|-----------|
| **Joint** | $81.2 \pm 0.2$ | $67.9 \pm 0.5$ | $78.4 \pm 0.4$ | $90.5 \pm 0.5$ | $84.7 \pm 0.3$ |
| **EM** | $\mathbf{83.6 \pm 0.3}$ | $\mathbf{72.7 \pm 0.5}$ | $\mathbf{79.6 \pm 0.5}$ | $\mathbf{92.2 \pm 0.3}$ | $\mathbf{86.5 \pm 0.2}$ |

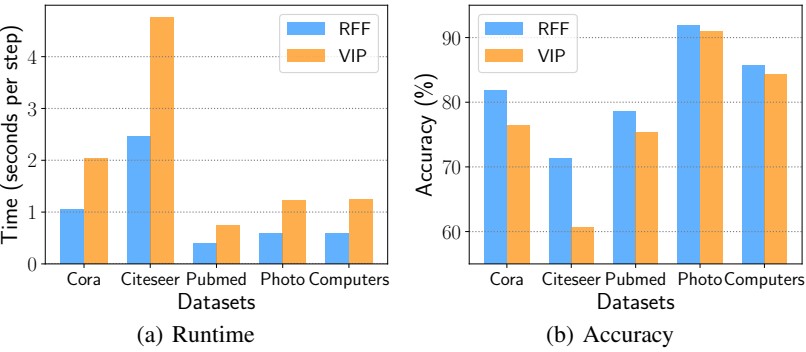

(a) Runtime      (b) Accuracy

Figure 2: (a) Training runtime comparison between proposed random Fourier features method (**RFF**) and variational inducing point method (**VIP**) with a batch size of $512$ on each dataset. (b) Classification accuracies of the proposed RFF method and VIP method on each dataset.

Table 6: Classification accuracies (%) of GPSRF and the baselines under different proportion of training edges on Cora and Photo, respectively.

| | Cora | | | | Photo | | | |
|---|---|---|---|---|---|---|---|---|
| | 0.1 | 0.3 | 0.5 | 0.7 | 0.1 | 0.3 | 0.5 | 0.7 |
| **GGP** | $\mathbf{66.8 \pm 0.4}$ | $71.9 \pm 0.5$ | $75.5 \pm 0.5$ | $79.7 \pm 0.4$ | $82.3 \pm 0.4$ | $84.4 \pm 0.5$ | $88.9 \pm 0.3$ | $89.3 \pm 0.3$ |
| **GCN** | $60.3 \pm 0.3$ | $69.4 \pm 0.5$ | $72.8 \pm 0.4$ | $77.1 \pm 0.3$ | $\mathbf{84.3 \pm 0.6}$ | $86.7 \pm 0.3$ | $87.8 \pm 0.4$ | $88.9 \pm 0.2$ |
| **GAT** | $63.5 \pm 0.2$ | $\mathbf{74.3 \pm 0.3}$ | $76.0 \pm 0.1$ | $79.3 \pm 0.3$ | $60.5 \pm 0.8$ | $80.5 \pm 0.6$ | $82.7 \pm 0.5$ | $83.9 \pm 0.5$ |
| **GPSRF** | $59.2 \pm 0.8$ | $72.1 \pm 0.6$ | $\mathbf{79.3 \pm 0.4}$ | $\mathbf{82.3 \pm 0.2}$ | $77.0 \pm 0.7$ | $\mathbf{87.1 \pm 0.1}$ | $\mathbf{89.6 \pm 0.4}$ | $\mathbf{90.8 \pm 0.5}$ |

## E.6 Runtime Comparison

In GPSRF, we use the random Fourier features of the proposed SRF kernel for scalable optimization. To verify that the proposed RFF method is more scalable than the variational inducing point methods, we further conduct experiments to compare their training runtimes. Specifically, in our GPSRF, we set the spectral distribution in SRF kernel as the Gaussian, such that the resulting kernel function is a RBF kernel. For comparison, we take GP-RBF with variational inducing point method [8] as baseline. Similar to GPSRF, in GP-RBF, we also use the same GraphSAGE network to learn representations. We use 200 inducing points in our experiments as we found it is the least number of inducing points to achieve satisfactory performance. We run the experiments on a single machine with 14 Intel Xeon CPUs (E5-2680 v4 @ 2.40GHz) and 220Gb of RAM. The experimental results in Figure 2 show that our GPSRF is not only faster than the variational inducing point method but also achieve better classification performance.

## E.7 The Effect of the Edge Sparsity

We further conduct experiments to study the effect of edge sparsity on the performance of our method. Specifically, we randomly remove different proportions of edges on two graph datasets, i.e., Cora and Photo, and then examine the performance of GPSRF and baselines on these sparsity graphs. The experimental results in Table 6 show the semi-supervised object classification performance. It's shown that our GPSRF can achieve the best performance when the proportion of training edges is relatively high. The results suggest that our GPSRF can perform better on dense graphs.