# OpenReview forum: "Structure-Aware Random Fourier Kernel for Graphs"
_NeurIPS.cc/2021/Conference — NeurIPS 2021 Poster_

### Official Review · Reviewer_Trb2 · 2021-07-14

**Rating:** 7
**Confidence:** 3

**Summary:**

The idea of this paper is to build a kernel on the features of the graph that takes into account the underlying structure of the graph. Once found, this kernel is used to define a GP on the graph. This is useful for learning tasks on graphs such as node classification/link prediction. In order to build this kernel, the authors propose to use a base translation invariant kernel that admits, by Bochner, a feature expansion, i.e. an embedding $\phi(x,w)$ (like RFF). More precisely, they propose to learn the law on the drawing of the frequencies $w$ of these features under the constraint that the features must incorporate the structure of the graph. These features are built in two steps: 1) the law on the drawing of frequencies $w$ is defined by a "hierarchical" law parameterized by a neural network that can be "arbitrarily complex" (in the manner of a GAN generator) 2) the features take the embedding of a subgraph centered at this node as input $x$. This embedding is defined by a GNN.

**Limitations And Societal Impact:**

Authors did not provide any potential negative societal impact of their work however, it is difficult for me to foresee these impacts as the work is mostly a methodological paper. There are some other limitations that I think should be more discussed. Especially concerning the fact that one needs to have a good GNN at first (see my review above).

**Main Review:**

- Pros:
	- The article is well written, quite clear.
	- The method proposed is interesting, and, to the best of my knowledge quite new.

- Cons:
	- The method supposes to have already a good embedding of the graph, which is rather limiting.
	- Some points in the experiments are not very clear.
	- The advantages and limitations of the approach could be better underlined.

Overall I find this article interesting, it combines many ideas and domains (kernels/GNN/GP) to define an interesting method for learning on structured data. I also find it well written, pedagogical, and I thank the authors for this effort. The proposed method seems to me quite relevant. There are however some points that are not very clear to me. Here are my detailed comments.

- About the method:

I find the method well detailed and clear. I think, the use of GNN embedding is the central point of the method. In this context, a point that seems unclear to me is that, as far as I understand, this embedding is not learned in conjunction with the kernel and is considered as fixed. Overall this implies to have already a good embedding of the graph, which can cause some problems in practice. Indeed, for the method to work, one needs to: 1) either have a good pre-trained GNN 2) or re-train a GNN on the data. Moreover, as shown in experiment 8.5 it seems that it is quite crucial to have a good embedding $g$. I find that this limitation is not discussed enough in the paper, especially in terms of learning complexity (section 7). It would have been good to point out that this complexity also depends on having a good GNN. On the other hand the method is flexible enough to consider more hand-crafted and unsupervised embeddings based on graph kernels for example. Did the authors test with other embeddings that are less dependent on a GNN? Moreover I think that ref [1] can be interesting in this context. The Variational EM part is quite "classical" for this kind of problems and seems relevant to me.

- About the experiments:

	- Semi supervised classification task. I am a bit skeptical about the message brought by this experiment. I find that the conclusion that "GPSRF > all GP models" is a bit misleading. As much as it is clear for most of the GPs, it is a bit less clear for the GPP method [2] (ref [17] of the article) which is the most similar to the method proposed in this article. The major difference between the two methods is that the base kernel of the graph-based GP of [2] is based on the Laplacian of the graph. So it is really interesting to compare with GPP. However, the performances between the two methods are very close, because of the variance, on all datasets (except maybe Computers). Especially since the performances are only calculated on one train/validation/test split. For me it is difficult to see that the difference between the two is statistically significant.
	- Link prediction. The results of this experiment seem convincing and show the advantage of the method on this type of task.
	- Regression task. It is not really clear to me the relevance of this experiment. More precisely I don't understand the interest of an experiment that considers only the base kernel (without the graph part). Is it to illustrate that the hierarchical Bayesian approach to learn the RF law is interesting? In this case, I think it would have been interesting to discuss a little bit why this idea is new (because I think that some works approach the same idea of learning the law of frequencies according to the data [3,4,5]).

One point that I thought was worth highlighting is the fact that the method in question can take into account/model uncertainty in the prediction. I think that this property is interesting and should be emphasized, because, to the best of my knowledge, it is rarely taken into account in the context of graphs.

- Minors:

I think it should be mitigated that methods relying on traditional kernels and Laplacian are wether "inefficient for them to capture the structural smoothness in graph data" (row 31) and "they ignore the rich features of objects and may not be capable to model complicated functions over large-scale graphs" (row 36). It is precisely what they are attempting to do, based on the prior that "close" nodes (in the topology of the graph) should share similar features, and, as always, the fact that these methods sometimes struggle with the learning task depends on the dataset.


[1] Daniele Zambon, Cesare Alippi, Lorenzo Livi. Graph Random Neural Features for Distance-Preserving Graph Representations

[2] Yin Cheng Ng, Nicolo Colombo, Ricardo Silva. Bayesian Semi-supervised Learning with Graph Gaussian Processes

[3] Zichao Yang, Alexander J. Smola, Le Song, Andrew Gordon Wilson. A la Carte - Learning Fast Kernels

[4] Aman Sinha, John C. Duchi. Learning Kernels with Random Features

[5] Junier Oliva, Avinava Dubey, Andrew G. Wilson, Barnabas Poczos, Jeff Schneider, Eric P. Xing. Bayesian Nonparametric Kernel-Learning


---- AFTER REBUTTAL ----

The authors addressed most of my concerns (see my comment below). I am changing my score to 7.


**Time Spent Reviewing:**

5

---

> ### Author Response · Authors · 2021-08-08
> **Response**
>
> We thank the reviewer for the detailed comments and suggestions, which is helpful for the improvement of the paper. The following is our point-to-point response to the reviewer’s concerns and comments. We believe that the issues mentioned in the comments will be properly addressed in the final version.
>
> 1 About the method:
>
> 1a) In our method, we actually do not require a pre-trained GNN since the GNN network is learned in conjunction with the kernel parameters. For details about the training procedure of our method, please see the pseudo code in Algorithm 1 of Appendix C. It can be seen from lines 7 and 12 of Algorithm 1 that the GNN $g$ is jointly learned with the kernel parameters. Such a joint learning can make our method more adaptive to the training data and obtain better performance. We admit that this point might not have been clearly stated in the main text and we thank the reviewer for pointing this out. We will clearly state the training procedure in the main text of our final version.
>
> Moreover, when comparing the computational complexity between our method and the variational inducing point method, we actually omit the computational complexity of GNN for clarity since both methods use the same GNN to learn node embeddings. Please see the experimental settings in Appendix E.6 for details about two methods. We will also clarify this point in our final version.
>
> 2a) We agree that our method is flexible enough to consider other node embedding techniques and we have actually tried different embedding methods in our experiments. Specifically, we have tried using summation, averaging, pooling operations [1], graph kernel method [2] and GNN to learn the node embeddings, and found that GNN performs best in our experiments. This is the reason why we use GNN as our aggregation function $g$. We also thank the reviewer for pointing out a relevant reference that might improve the performance of our method. We will conduct further experiments to examine its performance and report the results in the final version.
>
> 2 About the experiments:
>
> 2a) In the standard split of datasets, the results in Table 1 of the main text show that GPSRF performs better than GGP (sorry for the typo “GPP” in Table 1) on three out of five datasets (Citeseer, Pubmed and Computers) and achieve quite similar results with GGP on the other two datasets. We agree that the statement “GPSRF performs better than all the GP models” is a bit inaccurate. We will replace them with the above more precise statements in the final version. Moreover, we also report the classification results of GPSRF and baselines under different percentages of labeled data in Appendix E.1. The experimental results in Figure 1 of Appendix show that GPSRF can outperform GGP on Cora and Photo datasets under different label ratios. It would be very appreciated if the reviewer could take a look at this results in our Appendixes. Thank you very much!
>
> 2b) We propose a novel random Fourier (RF) kernel, which is defined by the spectral distribution (see Section 5.1 for details), and it is also the foundation for the proposed SRF kernel for graph data. The point of regression tasks is to empirically validate that the proposed RF kernel is an effective kernel function and is more expressive than the other kernel functions (RBF kernel and deep kernel). For the differences with previous works, it would be very appreciated if the reviewer is able to see the first point of our responses to reviewer TzfG. We will discuss the novelty of the RF kernel more thoroughly in the final version.
>
> 3 Thanks for the suggestion about the prediction uncertainties. We will elaborate more about this point in the main text and empirically show the utility of this property in the final version.
>
> 4 Minors:
> Thanks for the suggestion. We will modify these statements with some explanatory sentences in the final version.
>
>
> [1] Ying, Rex, Jiaxuan You, Christopher Morris, Xiang Ren, William L. Hamilton, and Jure Leskovec. Hierarchical graph representation learning with differentiable pooling.
>
> [2] Matteo Togninalli, Elisabetta Ghisu, Felipe Llinares-López, Bastian Rieck, and Karsten Borgwardt. Wasserstein Weisfeiler-Lehman graph kernels.

---

> > ### Comment · Reviewer_Trb2 · 2021-08-17
> > **Response to author's rebuttal**
> >
> > I would like to thank the authors for their response. I think the authors addressed most of my comments. Especially, I recognize that I did not understand that the GCN is learned in conjuction with the method, this is quite interesting. I think this point should be more highlighted.I also think it would be helpful to add Figure 1 of the supplement to the main text in the final version if the paper is accepted. Overall after reading the rebuttal I think this proposes an interesting approach and is technically sounded so that I am changing my score accordingly.

---

### Official Review · Reviewer_hdfy · 2021-07-16

**Rating:** 7
**Confidence:** 4

**Summary:**

This paper proposed a novel SRF kernel defined with spectral distribution based on the Fourier duality given by Bochner's theorem on the vector embedding of subgraphs using GCNN.  Based on the SRF kernel, the paper  proposes a GP model, GPSRF,  for graph learning tasks. Paper developed a variational EM algorithm to solve kernel parameters. It also studies the run time complexity of the proposed method. Model has been evaluated for semi-supervised object classification and link prediction. Experimental results show that GPSRF yields best performance compared with state-of-the-art baselines.

**Limitations And Societal Impact:**

Author has already discussed it in paper.

**Main Review:**

Use of GCNN to define a graph based kernel is new to me.
Combining subgraph embedding with Fourier kennel is interesting and might be very useful for many graph based tasks.
Advantage of this method is that it can capture features and graph structure in one kernel and the run time complexity is very less O(N).
Paper is well written and has clarity and is technically sound.
Experiment section is well planned and has been compared with many possible state of the art methods.
Paper has also discussed the possible negative social impact.

This model is applicable when an object is in a graph. Does it also applicable for tasks when the data object is a graph like protein classification etc?

Following related work is missing.
Nikolentzos, G., Meladianos, P., Tixier, A.J.P., Skianis, K. and Vazirgiannis, M., 2018, October. Kernel graph convolutional neural networks. In International conference on artificial neural networks (pp. 22-32). Springer, Cham.


**Time Spent Reviewing:**

5

---

> ### Author Response · Authors · 2021-08-08
> **Response**
>
> We thank the reviewer for the positive comments and the insightful question. The following is our response to the reviewer’s question about graph classification.
>
> We think our method is also applicable to graph classification tasks. In our SRF kernel, if we replace the local subgraphs around nodes with the graph objects, such as the protein networks, the SRF kernel can be used to calculate the similarity between two graph objects. As a result, we can carry out graph classification tasks in a similar manner as our object classification tasks.
>
> Thanks for pointing out the missing related work (KCNN), which is an important reference for our paper. Our method differs from KCNN in a number of ways: (1) KCNN uses graph kernels to learn graph representations to solve graph classification tasks. In contrast, our GPSRF uses graph neural network (GNN) to aggregate subgraph information to solve the node-oriented tasks such as node classification and link prediction. (2) KCNN uses convolution network (CNN) as the base model to make predictions while we leverage Gaussian Process as the base model. We will cite the paper and discuss these differences in the final version.

---

> > ### Comment · Reviewer_hdfy · 2021-08-24
> > **Thanks**
> >
> > I thank the author for their reply. Author's response is satisfactory to me, I would like to keep my score as it is.

---

### Official Review · Reviewer_TzfG · 2021-07-17

**Rating:** 6
**Confidence:** 4

**Summary:**


This paper develops a structure-aware random feature kernel for Gaussian process-based learning over graphs. Experiments were carried out on real-world datasets for classification and link prediction.


**Limitations And Societal Impact:**

The reviewer finds this aspect inadequate, while the authors discussed the limitation: require off-line training and not robust to attacks from adversary and claim that as negative societal impact, but it seems to the reviewer believe these are just algorithmic limitation instead of societal impact. The reviewer believes societal impact could be discussed regarding the potential privacy concern, as a lot of social network datasets are used which naturally leads to privacy concerns.

**Main Review:**

The reviewer finds the novelty of the proposed framework not enough. It is a direct combination of Gaussian process +Fourier Random Feature +GNN (GP+RF+GNN). However, GP+RF has already been studied in previous work, and the incorporation of graph structure is just by input the graph with the existing GNN framework to obtain a representation.

Moreover, in the experiments, it is shown that a threshold needs to be introduced for the proposed algorithm to obtain better results than GAT. However, the choice of \tau was not discussed. Moreover, the choice of some critical parameters such as L (hops of neighbors), F (number of RFs) is not discussed for the experiments either.

The reviewer finds the organization of the paper can be improved. Some important information is missing. For example, it is claimed that the output of g() is a subgraph representation obtained using GNN in (2) and (3), but (2) (3) only provides node-level representation. It is not clear how the subgraph representation is obtained.

The writing of the paper could be improved by clearly defining operators, discussing or clarifying the procedure of parameter selection, as well as clarifying the novelty compared with existing works.




**Time Spent Reviewing:**

6

---

> ### Author Response · Authors · 2021-08-08
> **Response**
>
> We thank the reviewer for the detailed comments. The following is our responses to the questions mentioned in the comments.
>
> 1 We can not agree that our method is a simple combination of existing methods, reasons of which are as follows:
> Firstly, although the random Fourier features (RFF) have been applied to Gaussian Process (GP) to approximate the pre-defined kernel function [1,2,3], these applications of RFF require calculating the spectral distribution of the pre-defined kernel function through the Fourier transform (see [1] or Appendix A for more details about RFF). We emphasize that for most complex kernel functions, it is not feasible to obtain the corresponding spectral distributions due to the intractability of Fourier transform. This is also the reason why RFF is only applicable to a small range of simple kernel functions so far, such as RBF kernel and arc-cosine kernel. In contrast, our random Fourier (RF) kernel is directly defined with a spectral distribution (we define the RF kernel with a hierarchy Bayesian distribution in our paper) based on Bochner's theorem. The definition of the RF kernel sidesteps calculating the potentially intractable spectral distribution. Also, we can obtain a complex and expressive kernel by defining a complicated spectral distribution. As a result, our method at least generalizes RFF to a family of complex and expressive kernel functions, i.e., RF kernels. For more details about the differences between our method and previous works, please see Line 132-149 of our main text and Appendix A.
>
> Secondly, the central idea of the proposed structure-aware random Fourier (SRF) kernel is to leverage the local subgraph structure around each node to capture the structural information. Such structure-aware kernels can improve the performance of GPs on graph data. To the best of our knowledge, the idea of leveraging local subgraph structure to define the kernel function between nodes has not been explored before. Moreover, we would like to point out that our method is a general framework since we can use various methods to aggregate the local subgraph information. We use GNN in our paper because it performs the best in our experiments due to its appropriate learning biases and strong representation capacity in real-world graphs. We can also use summation, averaging, pooling operations, or graph kernels [4] to aggregate the subgraph information. We will clarify the above points and report the experimental results of our model with different aggregation methods in our final version.
>
> 2 In fact, we have provided how we choose the hyperparameters, including $\tau$, in the Appendixes and discussed the impact of the choices of some critical parameters, such as hops of neighbors and number of RFs on the performance of our model in our paper. Specifically, we show how we choose the hyperparameter $\tau$ and the other hyperparameters for object classification tasks in Appendix D.4 but not the main body of the paper. We discuss the effects of the subgraph size $L$ in Appendix E.3. The number of RFs is actually the number of samples drawn from the spectral distribution and we denote it as $M$ in our paper instead of $F$ (dimension of input features). We discuss the effects of the number of samples $M$ in Section 8.4 of the main text. It would be very appreciated if you could take a look at our Appendixes. Thank you very much!
>
> 3 The goal of function $g( \mathcal{G}_i^L )$ is to aggregate the local subgraph information of node $v_i$. The output node-level representation of node $v_i$ given by GNN has already aggregated the subgraph information, so we only use the node-level representation of $v_i$ as the output of function $g$ in our method instead of the whole subgraph representation. We will clarify this point in our final version.
>
> 4 We will improve the writing of the paper based on the suggestions.
>
>
>
> [1] Ali Rahimi and Benjamin Recht. Random features for large-scale kernel machines.
>
> [2] Kurt Cutajar, Edwin V Bonilla, Pietro Michiardi, and Maurizio Filippone. Random feature expansions for deep gaussian processes.
>
> [3] Xiantong Zhen, Haoliang Sun, Yingjun Du, Jun Xu, Yilong Yin, Ling Shao, and Cees Snoek. Learning to learn kernels with variational random features.
>
> [4] Matteo Togninalli, Elisabetta Ghisu, Felipe Llinares-López, Bastian Rieck, and Karsten Borgwardt. Wasserstein Weisfeiler-Lehman graph kernels.

---

> > ### Comment · Reviewer_TzfG · 2021-08-17
> > **Response to authors' rebuttal**
> >
> > The reviewer appreciates the authors' response, and most of the concerns of the reviewer have been addressed.
> >
> > The reviewer agrees with the authors' response and acknowledges the novelty of the paper compared with existing GP+RF methods. The reviewer would suggest changing the name of RF kernels, as it is too similar to the naming in reference [1] (Random features for large-scale kernel machines) in the authors' rebuttal, while in fact, as the authors correctly pointed out, the RF kernel is one of the novelties that the reviewer missed.  The reviewer also agrees with the response that parameter selection has indeed been discussed in the appendix, which the authors may want to mention and direct the readers in the main text clearly.  Given this, the reviewer would like to raise the score.
> >
> > The reviewer however disagrees with the comment that "leveraging local subgraph structure to define the kernel function between nodes has not been explored before"  See for example survey paper [R1]  Page 7 section Neighborhood aggregation approaches "One of the dominating paradigms in the design of graph kernels is representation and comparison of local structure." In this section, several approaches based on local structure have been mentioned, e.g. [R2, R3]. See also R1-Page 13 section: subgraph patterns.  Since the local structure awareness is considered the "central idea" of the paper, the reviewer thinks a discussion and comparison of existing substructure-aware methods are very necessary and important and is missed.  The reviewer is however open to further raise the score if this could be clarified.
> >
> >
> > [R1] Kriege, N. M., Johansson, F. D., & Morris, C. (2020). A survey on graph kernels. Applied Network Science, 5(1), 1-42.
> >
> > [R2]Shervashidze N, Schweitzer P, van Leeuwen EJ, Mehlhorn K, Borgwardt KM (2011) Weisfeiler-Lehman graph kernels. J
> > Mach Learn Res 12:2539–2561
> >
> > [R3]Hido S, Kashima H (2009) A linear-time graph kernel. In: IEEE International Conference on Data Mining. pp 179–188.

---

> > > ### Author Response · Authors · 2021-08-18
> > > **Clarification of the differences with graph kernels**
> > >
> > > We thank the reviewer very much for the positive feedback and the detailed comments.
> > >
> > > We agree that leveraging local subgraph patterns has been previously explored in the design of graph kernels. However, we would like to highlight that the problem studied in this paper lies in a different domain than the graph kernels. We have discussed two major differences between our method and the graph kernels in Line 68-71 of the related work section (Section 2). (1) The proposed SGK kernel measures similarities between two nodes in a single graph, and the graph kernels measure similarities between two graphs. (2) The SGK kernel is proposed for the node-oriented tasks such as object classification and link prediction. In contrast, the graph kernels are proposed for the graph-level tasks such as graph classification. As a result, prior graph kernels such as random walk kernels [1] and Weisfeiler-Lehman kernels [2, 3] can not be directly applied in our research problem.
> > >
> > > Moreover, we also agree that it will be beneficial to discuss and compare the differences between the proposed SGK kernel and the substructure-aware graph kernels, despite the fact that they study problems in different domains. Firstly, the neighborhood-aggregation-based graph kernels [2, 4] aggregate the neighborhood’s label information while our SGK kernel aggregates the neighborhood information based on the continuous feature vectors. In our semi-supervised setting, where the labeled data is scarce, it is infeasible to straightforwardly apply these graph kernels to our problem. Secondly, the subgraph-based graph kernels [5, 6] propose to measure the similarities between two graphs by counting the occurrences of subgraph patterns (graphlets) of fixed size. Please note that these graph kernels, which are obtained based only on graph structures, do not consider any labels or feature vectors. It’s also computationally expensive to find all the subgraph patterns (graphlets) in the graphs. In contrast, our SGK kernel measures similarities between nodes by aggregating the L-hop neighborhood’s feature information using the aggregation function $g$, and it is also more computationally efficient to obtain these subgraph representations. We will elaborate the above differences between our SGK kernel and the graph kernels in our final version.
> > >
> > >
> > > [1] Gärtner, T., Flach, P., & Wrobel, S. (2003). On graph kernels: Hardness results and efficient alternatives. In Learning theory and kernel machines (pp. 129-143). Springer, Berlin, Heidelberg.
> > >
> > > [2] Shervashidze, N., Schweitzer, P., Van Leeuwen, E. J., Mehlhorn, K., & Borgwardt, K. M. (2011). Weisfeiler-Lehman graph kernels. Journal of Machine Learning Research, 12(9).
> > >
> > > [3] Schulz, T. H., Horváth, T., Welke, P., & Wrobel, S. (2021). A generalized weisfeiler-lehman graph kernel. arXiv preprint arXiv:2101.08104.
> > >
> > > [4] Hido, S., & Kashima, H. (2009). A linear-time graph kernel. In 2009 Ninth IEEE International Conference on Data Mining (pp. 179-188). IEEE.
> > >
> > > [5] Shervashidze, N., Vishwanathan, S. V. N., Petri, T., Mehlhorn, K., & Borgwardt, K. (2009). Efficient graphlet kernels for large graph comparison. In Artificial intelligence and statistics (pp. 488-495). PMLR.
> > >
> > > [6] Costa, F., & De Grave, K. (2010). Fast neighborhood subgraph pairwise distance kernel. In ICML.

---

> > > > ### Comment · Reviewer_TzfG · 2021-08-23
> > > > **Regarding graph kernels**
> > > >
> > > > The reviewer appreciates the authors' clarification. However, the reviewer does not fully agree with this rebuttal.
> > > >
> > > > While it is true most of the graph kernel methods are proposed for "graph-level tasks such as graph classification". The reviewer does not agree that existing methods cannot be used for node-level classification, or existing graph kernels do not cope with node-level similarity, and the problem considered here is in a "different domain".  As graph level is often considered an aggregation of node-level problems in the graphs.
> > > >
> > > > See e.g.,[1] [2] where the graph kernel is obtained by "comparing all pairs of nodes from graphs by iteratively comparing their neighborhoods". See also equations (1) in both [1,2]. where the graph kernel is defined by directly summing up all pairs of nodal kernels in graphs. In this sense, the node level kernel is a sub-component/sub-problem of the graph kernels.
> > > >
> > > > To be more specific, the node classification problem has actually be considered via developing a k-hop neighborhood-based kernel method in [1] for node representation and node classification tasks. where a function aggregating k-hop neighbor information is first used to extract nodal features, and a kernel function is applied to the obtained feature to characterize node level-similarity, which is very similar to the local-structure aware part in the current submission.
> > > >
> > > > Given the above reasons, the reviewer still believes the local structure-awareness for nodal similarity cannot be considered as a new problem in a "different domain" compared with graph kernels and suggests making a more careful claim on the contribution.
> > > >
> > > > Despite the disagreement with this rebuttal, the reviewer does agree with the 1st round rebuttal that this work "at least generalizes RFF to a family of complex and expressive kernel functions", and thinks this paper is worth publishing if the key contribution can be stated clearly, and the rating has been updated.
> > > >
> > > > [1] Tian, Y., Zhao, L., Peng, X., & Metaxas, D. (2019). Rethinking kernel methods for node representation learning on graphs. Advances in neural information processing systems, 32, 11686-11697.
> > > >
> > > > [2] Shervashidze, Nino, and Karsten M. Borgwardt. "Fast subtree kernels on graphs." Advances in neural information processing systems. 2009.

---

> > > > > ### Author Response · Authors · 2021-08-24
> > > > > **Response**
> > > > >
> > > > > We really appreciate the reviewer’s detailed and insightful comments as well as the constructive suggestions.
> > > > >
> > > > > We agree that node-level similarity is also an important problem in graph kernels and has been previously studied in some graph kernels. We will rephrase our contributions very carefully with more emphasis on the random Fourier (RF) kernels based on the suggestion. Moreover, we will also discuss and compare the differences between our method and existing substructure-aware graph kernels (please see the third paragraph of our second-round response) in our final version.

---

### Official Review · Reviewer_JdRE · 2021-07-18

**Rating:** 9
**Confidence:** 4

**Summary:**

Paper presents a novel graph-structure aware kernel for graph data that leverages local sub-graph structure for node label and link prediction. It leverages spectral representation of kernels and RFFs along with ideas from graph learning to present a variational EM algorithm for graph learning. Strong empirical results show efficacy of proposed method over non-GP based GNN learning approaches.

**Ethical Concerns:**

No ethics review concerns.

**Limitations And Societal Impact:**

Paper addresses some limitations of the graph learning framework in the conclusion but without any empirical results. Appendix E.1 addresses effect of label data imbalance. No direct societal impact of concern.

**Main Review:**

A key contribution of the paper is the idea of using local sub-graph structure around each node in order to define a structure-ware Bayesian graph kernel. It combines several ideas of spectral kernel representation, RFFs and GP's to proposes an interesting idea that leverages local neighborhood of a node for graph learning.

Paper is well written and technically strong and addresses several aspects of proposed framework including choice of distributions, convergence, label distribution impact, sample size, effect thresholding and so on. It also provides a Bayesian and computationally efficient algorithm to estimate model parameters.

Another aspect of Graph learning that is not considered in this work is most real-world graphs (random graphs with large N) are sparse with fewer edges and how does this impact EM modeling, convergence and results.

**Time Spent Reviewing:**

4

---

> ### Author Response · Authors · 2021-08-08
> **Response**
>
> We thank the reviewer for the positive comments and constructive suggestions. We agree that it is interesting to examine how the edge sparsity might affect the performance of our model. We will conduct further experiments on real-world graphs and random graphs with higher sparsity to study the impacts of the edge sparsity on the performance of our method in the final version, which can be done before the deadline of submitting the camera-ready paper.

---

### Decision · Program_Chairs · 2021-09-27

**Decision:**

Accept (Poster)

**Comment:**

The paper received positive reviews and the rebuttal phase clarified some potential
issues. Overall, this is a solid contribution, which should be accepted. The final submission
should, however, incorporate suggestions made during the rebuttal phase.